



# Multivariate hydrologic design methods under nonstationary conditions and application to engineering practice

Cong Jiang[1], Lihua Xiong[2], Lei Yan[2], Jianfan Dong[3] and Chong-Yu Xu[2,4]

[1]School of Environmental Studies, China University of Geosciences (Wuhan), Wuhan 430074, China

[2]State Key Laboratory of Water Resources and Hydropower Engineering Science, Wuhan University, Wuhan 430072, China

[3] Guangxi Water Resources Management Center, Nanning 530023, China

[4]Department of Geosciences, University of Oslo, P.O. Box 1047 Blindern, N-0316 Oslo, Norway

*Correspondence to*: Cong Jiang (jiangcong@cug.edu.cn)

**Abstract.** The multivariate hydrologic design under stationary condition is traditionally done through using the design criterion of return period, which theoretically equals to the average inter-arrival time of flood events divided by the exceedance probability of the design flood event. Under nonstationary conditions the exceedance probability of a given multivariate flood event would vary over time. This suggests that the traditional return period concept could not apply to the engineering practice under nonstationary conditions, since by such a definition a given multivariate flood event would correspond to a time-varying return period. In this paper, instead of return period, average annual reliability (AAR) is employed as the criterion for multivariate design, to ensure a given multivariate flood event would correspond to a unique design level under nonstationary conditions. The multivariate hydrologic design conditioned on the given ARR is estimated from the nonstationary multivariate flood distribution constructed by a dynamic C-vine copula, allowing for time-varying marginal distributions and dependence structure. Both the most-likely design event and confidence interval for the multivariate hydrologic design conditioned on the given AAR are identified to provide visual supporting information for designers. The multivariate flood series from the Xijiang River, China are chosen to perform a case study. The results indicate that both the marginal distributions and dependence structure of the multivariate flood series are nonstationary due to the driving force of urbanization and reservoir




regulation. The nonstationarities of both the marginal distributions and dependence structure can affect the outcome of the multivariate hydrologic design.

**Keywords:** Multivariate hydrologic design; Nonstationarity; Average annual reliability; Dynamic C-vine copula; Xijiang River



## 1. Introduction

A complete flood event or a flood hydrograph contain multiple feature variables, such as flood peak
and flood volume, and the safety of hydraulic structures could be associated with multiple flood
characteristics (Salvadori et al., 2007, 2011; Xiao et al., 2009; Xiong et al., 2015; Loveridge et al.,
2017). For example, the water level of a reservoir is controlled by not only flood peak flow but also
flood volume (Salvadori et al., 2011). Therefore, the multivariate hydrologic design, taking into account
multiple flood characteristics as well as their dependence, is able to provide a more rational design
strategy for hydraulic structures than the univariate hydrologic design (Zheng et al., 2013, 2014;
Balistrocchi and Bacchi, 2018).

The multivariate hydrologic design under stationary conditions has been widely investigated, and
the design criterion is usually quantified by return period ( $RP$ ), like for univariate hydrologic design.
Under the definition of the average recurrence interval between the flood events equaling or exceeding a
given threshold (Chow, 1964), the return period of a given flood event under stationary condition
theoretically equals to the average inter-arrival time between flood events divided by the exceedance
probability (Salvadori et al., 2011). On the other hand, the exceedance probability of a univariate flood
event is usually uniquely defined without ambiguity, while the exceedance probability of a multivariate
flood event could have multiple definitions (Salvadori et al., 2011; Vandenberghe et al., 2011). Up to
now, there are at least three kinds of definitions of the exceedance probability for a multivariate flood
event: the OR case that at least one of the flood features exceeds the prescribed threshold; the AND case
that all of the flood features exceed the prescribed thresholds; and the Kendall case that the univariate
representation transformed from the Kendall's distribution function exceeds the prescribed threshold
(Favre et al., 2004; Salvadori et al., 2007; Salvadori and De Michele, 2010; Vandenberghe et al., 2011).

Due to changing climatic conditions, as well as some anthropogenic driving force, such as land use
changes and river training, the nonstationarities of both univariate and multivariate flood series have
been widely reported (Xiong and Guo, 2004; Villarini et al., 2009; Vogel et al., 2011; López and
Francés, 2013; Bender et al., 2014; Xiong et al., 2015; Blöschl et al., 2017; Kundzewicz et al., 2018).
The multivariate flood distribution could exhibit more complex nonstationarity behaviors than the
univariate distribution, including the nonstationarities of the individual margins as well as the





nonstationarity of the dependence structure between the margins (Quessy et al., 2013; Bender et al., 2014; Xiong et al., 2015; Sarhadi et al., 2016; Kwon et al., 2016; Qi and Liu, 2017; Bracken et al., 2018). Both of the nonstationarities of the margins and dependence structure could impact the multivariate hydrologic designs. Under nonstationary conditions, the exceedance probability $p$ of a

given flood event would vary from year to year, and thus the return period, calculated as the average inter-arrival time between two successive flood events divided by $p$, is no longer a constant (Salas and Obeysekera, 2014; Jiang et al., 2015a; Sarhadi et al., 2016; Kwon et al., 2016; Yan et al., 2017). As a result, a given flood event would correspond to a time-varying and non-unique return period. Consequently, the traditional return period-based method for estimating hydrologic design may no

longer be applicable to the engineering practice under nonstationary conditions (Salas and Obeysekera, 2014).

In recent years, more and more attention has been paid to the hydrologic design under nonstationary conditions, but mainly focused on the univariate cases (Obeysekera and Salas, 2014; Obeysekera and Salas, 2016; Read and Vogel, 2016). In order to overcome the limitation of the traditional return period

under nonstationary conditions, the concept of return period has been revisited. Salas and Obeysekera (2014) extended two concepts of return period into a nonstationary framework, which are respectively defined as the expected waiting time (EWT) until an exceedance occurs (Olsen et al., 1998), and the time period that results in the expected number of exceedances (ENE) equal to one over this period (Parey et al., 2010).

Risk and reliability are both important measurements for assessing hydrologic designs (Vogel, 1987; Read and Vogel, 2015). Besides the redefinitions of return period, some risk-based or reliability-based metrics have been introduced as the hydrologic design criteria under nonstationary conditions (Rosner et al., 2014). Rootzén and Katz (2013) proposed the concept of design life level (DLL) to quantify hydrologic risk in a nonstationary climate during the whole design life period of hydraulic structures.

Read and Vogel (2015) introduced the concept of average annual reliability (AAR) to estimate the hydrologic design under nonstationary conditions. Liang et al. (2016) defined the equivalent reliability (ER) to estimate the design flood under nonstationary conditions by linking DLL to return period. These design criteria assess the risk or reliability of hydraulic structures associated with the flood distribution





during the whole design life period, rather than in a single year. For a given design life period, these

criteria can always yield a unique risk or reliability, and therefore they are applicable to the hydrologic designs under both stationary and nonstationary conditions. In general, these criteria could yield similar outcomes of hydrologic design (Yan et al., 2017).

Besides, under the multivariate framework a given design level would correspond to an infinite number of possible hydrologic design events (Hawkes, 2008; Kew et al., 2013; Fu et al., 2014; Zheng et

al., 2015, 2017), but these design events are generally not equivalent because their joint probability density values (i.e. likelihood) are usually different (Volpi and Fiori, 2012; Li et al., 2017; Yin et al., 2017). In engineering practice, it should be necessary to determine a typical design event as the representative for a specific design level. For example, in Chinese engineering practice a unique design flood hydrograph corresponding to a given design level is usually required to determine the scale of

hydraulic structures (Yin et al., 2017). The most-likely design event, which theoretically has the largest joint probability density (likelihood) among all possible design events (Salvadori et al., 2011), seems to be the best representative candidate. Besides the most-likely design event, the confidence interval for the countless possible design events is also necessarily identified to provide a finite design range for designers (Volpi and Fiori, 2012; Yin et al., 2017). The most-likely design event and confidence interval

for the bivariate hydrologic design under stationary condition have been identified (Salvadori et al., 2011; Volpi and Fiori, 2012; Li et al., 2017; Yin et al., 2017), but the cases for higher-dimensional hydrologic designs as well as under nonstationary conditions have not yet been covered in current studies.

Therefore, the objective of the study is to address the issue of multivariate hydrologic design

applying to the engineering practice under nonstationary conditions, which is achieved through the following steps. First, the nonstationary multivariate flood distribution is constructed using a dynamic canonical vine (C-vine) copula (Aas et al., 2009), which is able to capture the nonstationarities of both marginal distributions and dependence structure. Then, the design criterion for the multivariate flood event is quantified by average annual reliability (AAR) instead of traditional return period, since a given

multivariate flood event would correspond to a unique AAR under both stationary and nonstationary conditions (Read and Vogel, 2015; Yan et al., 2017). The multivariate hydrologic design for any given





AAR can be estimated from the nonstationary multivariate flood distribution.

The above described methods for the multivariate hydrologic design under nonstationary conditions are applied to the Xijiang River, China. The four-dimensional multivariate flood series, including the annual maximum daily discharge, annual maximum 3-day flood volume, annual maximum 7-day flood volume and annual maximum 15-day flood volume of the Xijiang River, are chosen as the study data, because they are the feature variables for deriving the design flood hydrograph for hydraulic structures. It has been found that the natural flood processes of this river have been significantly altered by urbanization and reservoir regulation (Xu et al., 2014), but these two factors have not yet been taken into account in the multivariate hydrologic design.

The remainder of this paper is organized as follows. The next section presents the methods developed in this paper. Section 3 describes the study area and data used in this paper. The results of the case study are given in section 4. Finally, the conclusion and discussion of this study are summarized in section 5.

## 2. Methods

This section presents the methods developed in this study, which consist of the following: (1) nonstationary multivariate flood distribution based on a dynamic C-vine copula, allowing for both time-varying marginal distributions and time-varying dependence structure; and (2) estimation of the multivariate hydrologic design associated with average annual reliability (AAR) under nonstationary conditions. The annual maximum daily discharge ($Q_1$), annual maximum 3-day flood volume ($V_3$), annual maximum 7-day flood volume ($V_7$) and annual maximum 15-day flood volume ($V_{15}$) are chosen to define the four-dimensional multivariate flood series $(Q_1, V_3, V_7, V_{15})$. In China, these four flood series are all key feature variables required for deriving the design flood hydrograph for hydraulic structures (Ministry of Water Resources of People's Republic of China, 1996; Xiao et al., 2009; Xiong et al., 2015; Li et al., 2017).

### 2.1. Probability distribution of the nonstationary multivariate flood series

According to Sklar's Theorem (Sklar, 1959), the probability distribution of the four-dimensional flood series $(Q_1, V_3, V_7, V_{15})$ at time t ($t = 1, 2, ..., n$, and $n$ is the length of the flood series) can be




formulated through a copula $C(\cdot)$ as follows

$$
\begin{aligned}
& F\left(Q_{1,t}, V_{3,t}, V_{7,t}, V_{15,t} \mid \boldsymbol{\theta}_t\right) \\
& = C\left[F_1\left(Q_{1,t} \mid \boldsymbol{\theta}_{1,t}\right), F_3\left(V_{3,t} \mid \boldsymbol{\theta}_{3,t}\right), F_7\left(V_{7,t} \mid \boldsymbol{\theta}_{7,t}\right), F_{15}\left(V_{15,t} \mid \boldsymbol{\theta}_{15,t}\right) \mid \boldsymbol{\theta}_{c,t}\right] \\
& = C\left[u_{1,t}, u_{3,t}, u_{7,t}, u_{15,t} \mid \boldsymbol{\theta}_{c,t}\right]
\end{aligned}
\tag{1}
$$

where $F_1\left(Q_{1,t} \mid \boldsymbol{\theta}_{1,t}\right)$, $F_3\left(V_{3,t} \mid \boldsymbol{\theta}_{3,t}\right)$, $F_7\left(V_{7,t} \mid \boldsymbol{\theta}_{7,t}\right)$ and $F_{15}\left(V_{15,t} \mid \boldsymbol{\theta}_{15,t}\right)$ denote the marginal distributions

for $Q_1$, $V_3$, $V_7$ and $V_{15}$, respectively; $u_{1,t}$, $u_{3,t}$, $u_{7,t}$ and $u_{15,t}$ are the marginal probabilities of $Q_1$, $V_3$, $V_7$

and $V_{15}$, respectively; $\boldsymbol{\theta}_{1,t}$, $\boldsymbol{\theta}_{3,t}$, $\boldsymbol{\theta}_{7,t}$ and $\boldsymbol{\theta}_{15,t}$ are the corresponding distribution parameters; and $\boldsymbol{\theta}_{c,t}$

stands for the copula parameter vector, which describes the strength of the dependence structure.

$\boldsymbol{\theta}_t = \left(\boldsymbol{\theta}_{1,t}, \boldsymbol{\theta}_{3,t}, \boldsymbol{\theta}_{7,t}, \boldsymbol{\theta}_{15,t}, \boldsymbol{\theta}_{c,t}\right)$ is the parameter vector of the whole multivariate distribution, including the

marginal distribution parameters as well as the copula parameters.

According to the multivariate distribution of $\left(Q_1, V_3, V_7, V_{15}\right)$ defined by Eq. (1), the corresponding

density function can be written as

$$
\begin{aligned}
& f\left(Q_{1,t}, V_{3,t}, V_{7,t}, V_{15,t} \mid \boldsymbol{\theta}_t\right) \\
& = c\left[F_1\left(Q_{1,t} \mid \boldsymbol{\theta}_{1,t}\right), F_3\left(V_{3,t} \mid \boldsymbol{\theta}_{3,t}\right), F_7\left(V_{7,t} \mid \boldsymbol{\theta}_{7,t}\right), F_{15}\left(V_{15,t} \mid \boldsymbol{\theta}_{15,t}\right) \mid \boldsymbol{\theta}_{c,t}\right] \cdot \\
& \quad f_1\left(Q_{1,t} \mid \boldsymbol{\theta}_{1,t}\right) \cdot f_3\left(V_{3,t} \mid \boldsymbol{\theta}_{3,t}\right) \cdot f_7\left(V_{7,t} \mid \boldsymbol{\theta}_{7,t}\right) \cdot f_{15}\left(V_{15,t} \mid \boldsymbol{\theta}_{15,t}\right) \\
& = c\left(u_{1,t}, u_{3,t}, u_{7,t}, u_{15,t} \mid \boldsymbol{\theta}_{c,t}\right) \cdot f_1\left(Q_{1,t} \mid \boldsymbol{\theta}_{1,t}\right) \cdot f_3\left(V_{3,t} \mid \boldsymbol{\theta}_{3,t}\right) \cdot f_7\left(V_{7,t} \mid \boldsymbol{\theta}_{7,t}\right) \cdot f_{15}\left(V_{15,t} \mid \boldsymbol{\theta}_{15,t}\right)
\end{aligned}
\tag{2}
$$

where $f_1\left(Q_{1,t} \mid \boldsymbol{\theta}_{1,t}\right)$, $f_3\left(V_{3,t} \mid \boldsymbol{\theta}_{3,t}\right)$, $f_7\left(V_{7,t} \mid \boldsymbol{\theta}_{7,t}\right)$ and $f_{15}\left(V_{15,t} \mid \boldsymbol{\theta}_{15,t}\right)$ are the density functions of the

marginal distributions for $Q_1$, $V_3$, $V_7$ and $V_{15}$, respectively. $c(\cdot)$ denotes the density function of copula

$C(\cdot)$. Seen from Eq. (2), the multivariate distribution of $\left(Q_1, V_3, V_7, V_{15}\right)$ can be separated into two

modules, including the marginal distributions, i.e., $f_1\left(Q_{1,t} \mid \boldsymbol{\theta}_{1,t}\right)$, $f_3\left(V_{3,t} \mid \boldsymbol{\theta}_{3,t}\right)$, $f_7\left(V_{7,t} \mid \boldsymbol{\theta}_{7,t}\right)$ and

$f_{15}\left(V_{15,t} \mid \boldsymbol{\theta}_{15,t}\right)$, as well as the dependence structure expressed by the copula density function

$c\left(u_{1,t}, u_{3,t}, u_{7,t}, u_{15,t} \mid \boldsymbol{\theta}_{c,t}\right)$. Under nonstationary conditions, both of the margins and dependence structure

of $\left(Q_1, V_3, V_7, V_{15}\right)$ could vary with time t. In the next two sub-sections, the details of modeling the time-





varying marginal distributions and time-varying dependence structure will be presented.

### 2.1.1. Nonstationary marginal distributions based on time-varying moments model

The time-varying moments model, in which the distribution parameters or moments are expressed as functions of time variable or some other explanatory variable(s), has been widely employed to capture the nonstationarity of univariate flood series (Strupczewski et al., 2001; Villarini et al., 2009). In this study, the nonstationary marginal distributions of the multivariate flood series $\left(Q_1, V_3, V_7, V_{15}\right)$ are constructed by the time-varying moments model. To ensure the location parameter $\mu$ (referring to the first moment or mean of flood series) strictly positive, $\mu$ is expressed as an exponential function of the covariates $\left(x_1, x_2, ..., x_k\right)$ as follows

$$\mu_t = \exp\left(\alpha_0 + \sum_{i=1}^{k} \alpha_i x_{i,t}\right) \tag{3}$$

where $\left(\alpha_0, \alpha_1, ..., \alpha_k\right)$ is the model parameter vector, and is estimated using the maximum likelihood estimate (MLE) method (Strupczewski et al., 2001). The higher-order distribution parameters such as scale and shape parameters are assumed to be kept constant to avoid possible larger uncertainty in parameter estimation, although they could also be nonstationary.

Five widely-used probability distributions in flood frequency analysis, including three-parameter Pearson type III (PIII) distribution and Generalized Extreme Value (GEV) distribution, and two-parameter Gamma distribution, Weibull distribution and Lognormal distribution, are employed as the candidate distributions for the univariate flood series (Villarini et al., 2009; Yan et al., 2017). The goodness of fit (GOF) of the probability distributions is examined by using the Kolmogorov-Smirnov (KS) test (Frank and Massey, 1951). The relative fitting quality of the time-varying moments models as well as the candidate distributions is assessed by the Akaike Information Criterion (AIC; Akaike, 1974). The best model featured with the smallest AIC value is chosen to describe the marginal distributions, from the nonstationary models generally expressed by Eq. (3).

### 2.1.2. Nonstationary dependence structure based on dynamic C-vine copula

After estimating the marginal distributions, the next step is to construct the nonstationary



dependence structure of $\left(Q_1,V_3,V_7,V_{15}\right)$ as formulated by the copula density function $c\left(u_{1,t},u_{3,t},u_{7,t},u_{15,t}\mid\boldsymbol{\theta}_{c,t}\right)$. Given that most applied copula functions are for bivariate random variables, $c\left(u_{1,t},u_{3,t},u_{7,t},u_{15,t}\mid\boldsymbol{\theta}_{c,t}\right)$ cannot be directly expressed as a specific copula function. The pair copula method has proved its powerful ability in constructing the distribution of multivariate random variables

through decomposing the multivariate probability density into a series of bivariate copulas (Aas et al., 2009; Xiong et al., 2015). In this study the dependence structure of $\left(Q_1,V_3,V_7,V_{15}\right)$ is constructed by the pair copula method.

Numerous pair-copula decomposition forms for a multivariate distribution are available, among which there are two kinds of decompositions with the regular vine structures named canonical vine (C-

vine) and drawable vine (D-vine) prevailing in practice. It is known that the C-vine is more suitable when there is a key variable governing multivariate dependence, while the D-vine more resembles independence graphs (Aas et al., 2009). In this study, the C-vine copula with $Q_1$ elected as the key variable is employed to construct the dependence structure of $\left(Q_1,V_3,V_7,V_{15}\right)$, since the flood peak $Q_1$ is usually regarded as the dominant feature for a flood hydrograph (Ministry of Water Resources of

People's Republic of China, 1996). Thus, the density function $c\left(u_{1,t},u_{3,t},u_{7,t},u_{15,t}\mid\boldsymbol{\theta}_{c,t}\right)$ can be decomposed into six bivariate pair copulas as follows

$$
\begin{aligned}
c\left(u_{1,t},u_{3,t},u_{7,t},u_{15,t}\mid\boldsymbol{\theta}_{c,t}\right) = {} & c_{13}\left(u_{1,t},u_{3,t}\mid\theta_{13,t}\right)\cdot c_{17}\left(u_{1,t},u_{7,t}\mid\theta_{17,t}\right)\cdot c_{115}\left(u_{1,t},u_{15,t}\mid\theta_{115,t}\right)\cdot \\
& c_{37\mid1}\left[F\left(u_{3,t}\mid u_{1,t}\right),F\left(u_{7,t}\mid u_{1,t}\right)\middle|\theta_{37\mid1,t}\right]\cdot \\
& c_{315\mid1}\left[F\left(u_{3,t}\mid u_{1,t}\right),F\left(u_{15,t}\mid u_{1,t}\right)\middle|\theta_{315\mid1,t}\right]\cdot \\
& c_{715\mid13}\left[F\left(u_{7,t}\mid u_{1,t},u_{3,t}\right),F\left(u_{15,t}\mid u_{1,t},u_{3,t}\right)\middle|\theta_{715\mid13,t}\right]
\end{aligned}
\tag{4}
$$

where $\boldsymbol{\theta}_{c,t}=\left(\theta_{13,t},\theta_{17,t},\theta_{115,t},\theta_{37\mid1,t},\theta_{315\mid1,t},\theta_{715\mid13,t}\right)$ is the parameter vector in the C-vine copula, and



$$F\left(u_{3,t} \mid u_{1,t}\right) = \frac{\partial C_{13}\left(u_{1,t}, u_{3,t} \mid \theta_{13,t}\right)}{\partial u_{1,t}}$$

$$F\left(u_{7,t} \mid u_{1,t}\right) = \frac{\partial C_{17}\left(u_{1,t}, u_{7,t} \mid \theta_{17,t}\right)}{\partial u_{1,t}}$$

$$F\left(u_{15,t} \mid u_{1,t}\right) = \frac{\partial C_{115}\left(u_{1,t}, u_{15,t} \mid \theta_{115,t}\right)}{\partial u_{1,t}} \qquad (5)$$

$$F\left(u_{7,t} \mid u_{1,t}, u_{3,t}\right) = \frac{\partial c_{37|1}\left[F\left(u_{3,t} \mid u_{1,t}\right), F\left(u_{7,t} \mid u_{1,t}\right) \mid \theta_{37|1,t}\right]}{\partial F\left(u_{3,t} \mid u_{1,t}\right)}$$

$$F\left(u_{15,t} \mid u_{1,t}, u_{3,t}\right) = \frac{\partial C_{315|1}\left[F\left(u_{3,t} \mid u_{1,t}\right), F\left(u_{15,t} \mid u_{1,t}\right) \mid \theta_{315|1,t}\right]}{\partial F\left(u_{3,t} \mid u_{1,t}\right)}$$

Figure 1 shows the schematic decomposition of the four-dimensional C-vine copula as expressed by Eq. (4). It can be seen that the four-dimensional C-vine copula has a hierarchical structure with three trees and six edges. The first tree (T1) includes three bivariate pair copulas, i.e., $c_{13}\left(\cdot \mid \theta_{13,t}\right)$, $c_{17}\left(\cdot \mid \theta_{17,t}\right)$ and $c_{115}\left(\cdot \mid \theta_{115,t}\right)$, which directly act on the marginal probabilities and actually describe the bivariate dependences between the key variable $Q_1$ with the other three variables, i.e., $V_3$, $V_7$ and $V_{15}$. The second

tree (T2) includes two bivariate pair copulas $c_{37|1}\left(\cdot \mid \theta_{37|1,t}\right)$ and $c_{315|1}\left(\cdot \mid \theta_{315|1,t}\right)$, which act on the conditional distribution functions with $u_{1,t}$ as the conditioning variable. Finally, the third tree (T3) includes only one bivariate pair copula $c_{715|13}\left(\cdot \mid \theta_{715|13,t}\right)$ acting on conditional distribution functions with both $u_{1,t}$ and $u_{3,t}$ as the conditioning variables.

Similar to the nonstationary marginal distributions, the nonstationarity of the dependence structure

of $\left(Q_1, V_3, V_7, V_{15}\right)$ is characterized by the time variation of copula parameter. In this study, we just consider the nonstationarities of the copula parameters in T1, i.e., $\theta_{13,t}$, $\theta_{17,t}$ and $\theta_{115,t}$, which actually measure the bivariate dependence of $\left(Q_1, V_3\right)$, $\left(Q_1, V_7\right)$ and $\left(Q_1, V_{15}\right)$, respectively. Here for the robust parameter estimation, the copula parameters $\theta_{37|1,t}$ and $\theta_{315|1,t}$ in T2 and as well as $\theta_{715|13,t}$ in T3 are kept constant, although they could be assumed to be time-varying like those parameters in T1.



Since the bivariate Gumbel-Hougaard copula accounts for the upper tail dependence and is well-suited to the dependence structure of multivariate flood distribution (Salvadori et al., 2007; Zhang and Singh, 2007), it will be employed to construct the dynamic C-vine copula formulated by Eq. (4). The bivariate Gumbel-Hougaard copula is expressed as follows

$$C(u,v) = \exp\left\{-\left[(-\ln u)^{\theta_c} + (-\ln v)^{\theta_c}\right]^{1/\theta_c}\right\}, \ \theta_c \in [1,\infty) \tag{6}$$

where $u$ and $v$ denote the bivariate marginal probabilities, and $\theta_c$ is the single parameter measuring the dependence strength. The time-varying copula parameter $\theta_c$ is generally expressed as an expression of the covariate vector $(x_1, x_2, ..., x_l)$ ($l$ is the number of the covariates) as follows

$$\theta_{c,t} = 1 + \exp\left(\beta_0 + \sum_{i=1}^{l} \beta_i x_{i,t}\right) \tag{7}$$

where $(\beta_0, \beta_1, \beta_2, ..., \beta_l)$ is the parameter vector and estimated by using the MLE method (Aas et al.,

2009). Here the copula parameter $\theta_c$ is written as the sum of one and an exponential function of the covariates to satisfy its domain range $\theta_c \in [1,\infty)$. The best nonstationary model for each pair copula in T1 is chosen from the nonstationary models generally expressed by Eq. (7) in terms of AIC value (Akaike, 1974). Then, the rest copula parameters $\theta_{37|1,t}$, $\theta_{315|1,t}$ and $\theta_{715|13,t}$ in T2 and T3 are estimated in sequence also by using the MLE method. The goodness of fit (GOF) of the C-vine copula is examined

by the Probability Integral Transform (PIT) test (Aas et al., 2009).

As described above, in this study the marginal and dependence parameters are separately estimated according to the Inference Function for Margins (IFM) method (Joe, 1997; Fu and Butler, 2014; Jiang et al., 2015a). It is worth noting that the marginal and dependence parameters can be simultaneously estimated by the conventional Maximum Likelihood (ML) method. But given that the nonstationary and

multivariate flood distribution constructed in this paper is very complex and includes numerous parameters, the IFM method should be the more robust way.

## 2.2. Multivariate hydrologic design under nonstationary conditions

### 2.2.1.  Average annual reliability for multivariate flood event




As the AAR introduced by Read and Vogel (2015) is calculated by the simple arithmetic average
method, it takes into account the annual nonstationary reliability of each year with the same weighting
factor. In terms of safer design strategy, worse (i.e. lower) annual reliability should deserve more
concerns, but this is something that the arithmetic-average AAR is not capable of doing. In this study,
the geometric average method, which is more dominated by the minimum than the arithmetic average is
and could theoretically yield safer design values, is employed to calculate AAR. Actually, the
geometric-average AAR is also equivalent to the metrics of DLL (Rootzén and Katz, 2013) and ER
(Liang et al., 2016)

Denote $(q_1, v_3, v_7, v_{15})$ as a given multivariate flood event, and the exceedance probability $p_t$ of
$(q_1, v_3, v_7, v_{15})$ would vary from year to year under nonstationary conditions. AAR for $(q_1, v_3, v_7, v_{15})$ is
calculated by the geometric average method as follows

$$AAR(q_1, v_3, v_7, v_{15}) = \left[ \prod_{t=T_1}^{T_2} (1 - p_t) \right]^{\frac{1}{T_2 - T_1 + 1}} \tag{8}$$

where $T_1$ and $T_2$ stand for the beginning year and ending year of the operation of an assumed hydraulic
structure respectively, and therefore $T_2 - T_1 + 1$ is the length of the design life period of the assumed
hydraulic structure. $1 - p_t$ measures the annual reliability for the given multivariate flood event
$(q_1, v_3, v_7, v_{15})$ at time $t$.

### 2.2.2. Exceedance probabilities of multivariate flood event

In this paper, we characterize AAR by considering three kinds of definitions for the exceedance
probabilities of the multivariate flood event $(q_1, v_3, v_7, v_{15})$, i.e. OR case, AND case and Kendall case
(Favre et al., 2004; Salvadori et al., 2007; Salvadori and De Michele, 2010; Vandenberghe et al., 2011).
The OR case for $(q_1, v_3, v_7, v_{15})$ is that at least one of the flood features exceeds the prescribed threshold.
The exceedance probability in the OR case at time $t$ is denoted as $p_t^{or}$, and calculated by

$$p_t^{or} = P(Q_{1,t} \geq q_1 \vee V_{3,t} \geq v_3 \vee V_{7,t} \geq v_7 \vee V_{15,t} \geq v_{15}) = 1 - F(q_1, v_3, v_7, v_{15} | \boldsymbol{\theta}_t) \tag{9}$$



where '$\vee$' stands for the OR operator, and $F\left(\cdot|\boldsymbol{\theta}_t\right)$ is defined in Eq. (1).

The AND case for $\left(q_1, v_3, v_7, v_{15}\right)$ is that all of the flood features exceed the prescribed thresholds, and the corresponding exceedance probability $p_t^{and}$ at time $t$ is

$$
\begin{aligned}
p_t^{and} &= P\left(Q_{1,t} > q_1 \wedge V_{3,t} > v_3 \wedge V_{7,t} > v_7 \wedge V_{15,t} > v_{15}\right) \\
&= \iiiint_{\Omega^{and}} f\left(Q_{1,t}, V_{3,t}, V_{7,t}, V_{15,t}|\boldsymbol{\theta}_t\right) dQ_{1,t} \cdot dV_{3,t} \cdot dV_{7,t} \cdot dV_{15,t} \\
\Omega^{and} &: q_1 < Q_{1,t} < \infty, v_3 < V_{3,t} < \infty, v_7 < V_{7,t} < \infty, v_{15} < V_{15,t} < \infty
\end{aligned}
\tag{10}
$$

where '$\wedge$' is the AND operator, and $f\left(\cdot|\boldsymbol{\theta}_t\right)$ is defined in Eq. (2).

As for the Kendall case, the multivariate flood event $\left(q_1, v_3, v_7, v_{15}\right)$ is first transformed into a univariate representation via the Kendall's distribution function $K_C\left(\cdot\right)$ as follows

$$
K_C\left(\rho_t\right) = P\left[C\left(u_{1,t}, u_{3,t}, u_{7,t}, u_{15,t}|\boldsymbol{\theta}_{c,t}\right) \le \rho_t\right] = P\left[F\left(Q_{1,t}, V_{3,t}, V_{7,t}, V_{15,t}|\boldsymbol{\theta}_t\right) \le \rho_t\right]
\tag{11}
$$

where $\rho_t = F\left(q_1, v_3, v_7, v_{15}|\boldsymbol{\theta}_t\right)$ is the probability level corresponding to the given flood event $\left(q_1, v_3, v_7, v_{15}\right)$. For the general multivariate case, the Kendall's distribution function cannot be analytically formulated as a specific expression, but can be numerically estimated through the Monte Carlo method (Niederreiter, 1978; Salvadori et al., 2011). Then, the corresponding exceedance probability $p_t^{ken}$ in the Kendall case at time $t$ is given as follows

$$
p_t^{ken} = 1 - K_C\left(\rho_t\right)
\tag{12}
$$

Replacing the exceedance probability $p_t$ in Eq. (8) by $p_t^{or}$, $p_t^{and}$ and $p_t^{ken}$, the average annual reliability in the OR, AND and Kendall cases can be respectively calculated.

### 2.2.3. Most-likely design event and confidence interval for multivariate hydrologic design

In this section, the methods identifying the most-like design event, denoted by $\left(z_{Q_1}^*, z_{V_3}^*, z_{V_7}^*, z_{V_{15}}^*\right)$, and confidence interval for the multivariate hydrologic design $\left(z_{Q_1}, z_{V_3}, z_{V_7}, z_{V_{15}}\right)$ corresponding to the given AAR (denoted by $\eta$) will be introduced. During the whole design life period from $T_1$ to $T_2$, the



average annual probability density, denoted by $g(\cdot)$, of the multivariate hydrologic design $\left(z_{Q_1}, z_{V_3}, z_{V_7}, z_{V_{15}}\right)$ is expressed as follows

$$g\left(z_{Q_1}, z_{V_3}, z_{V_7}, z_{V_{15}}\right) = \frac{1}{T_2 - T_1 + 1}\sum_{t=T_1}^{T_2} f\left(z_{Q_1}, z_{V_3}, z_{V_7}, z_{V_{15}} \big| \boldsymbol{\theta}_t\right) \tag{13}$$

The probability distribution function for $AAR \le \eta$ can be written as

$$\Phi(\eta) = \underset{\Omega: AAR(q_1, v_3, v_7, v_{15}) \le \eta}{\iiiint} g\left(q_1, v_3, v_7, v_{15}\right) dq_1 dv_3 dv_7 dv_{15} \tag{14}$$

Denoting the density function of $\Phi(\eta)$ as $\phi(\eta)$, the probability density of $\left(z_{Q_1}, z_{V_3}, z_{V_7}, z_{V_{15}}\right)$ conditioned on $AAR = \eta$ can be expressed as

$$g_{|AAR=\eta}\left(z_{Q_1}, z_{V_3}, z_{V_7}, z_{V_{15}}\right) = \frac{g\left(z_{Q_1}, z_{V_3}, z_{V_7}, z_{V_{15}}\right)}{\phi(\eta)} \tag{15}$$

And the most-likely design event conditioned on $AAR = \eta$ is theoretically given as follows

$$\left(z_{Q_1}^*, z_{V_3}^*, z_{V_7}^*, z_{V_{15}}^*\right) = \arg\max g_{|AAR=\eta}\left(z_{Q_1}, z_{V_3}, z_{V_7}, z_{V_{15}}\right) \tag{16}$$

Unfortunately, the analytical solutions of both the most-likely design event $\left(z_{Q_1}^*, z_{V_3}^*, z_{V_7}^*, z_{V_{15}}^*\right)$ and confidence interval are unavailable, but can be approximately estimated through the Monte Carlo simulation method. First, the design events with the sample size $N$ conditioned on $AAR = \eta$ are

generated. Then, these design events are sorted in descending order of their multivariate probability density, and denoted by

$$\left(z_{Q_1}^1, z_{V_3}^1, z_{V_7}^1, z_{V_{15}}^1\right), \left(z_{Q_1}^2, z_{V_3}^2, z_{V_7}^2, z_{V_{15}}^2\right), \cdots, \left(z_{Q_1}^{Nc}, z_{V_3}^{Nc}, z_{V_7}^{Nc}, z_{V_{15}}^{Nc}\right), \cdots, \left(z_{Q_1}^N, z_{V_3}^N, z_{V_7}^N, z_{V_{15}}^N\right) \tag{17}$$

where $Nc = N \cdot p_c$, and $p_c$ is the critical probability level for the confidence interval. Thus, the approximate solution for $\left(z_{Q_1}^*, z_{V_3}^*, z_{V_7}^*, z_{V_{15}}^*\right)$ is $\left(z_{Q_1}^1, z_{V_3}^1, z_{V_7}^1, z_{V_{15}}^1\right)$. The lower boundary for the confidence

interval is given as follows





$$
\begin{cases}
z_{Q_1}^L = \min\left(z_{Q_1}^1, z_{Q_1}^2, \cdots, z_{Q_1}^{Nc}\right) \\
z_{V_3}^L = \min\left(z_{V_3}^1, z_{V_3}^2, \cdots, z_{V_3}^{Nc}\right) \\
z_{V_7}^L = \min\left(z_{V_7}^1, z_{V_7}^2, \cdots, z_{V_7}^{Nc}\right) \\
z_{V_{15}}^L = \min\left(z_{V_{15}}^1, z_{V_{15}}^2, \cdots, z_{V_{15}}^{Nc}\right)
\end{cases}
\tag{18}
$$

The upper boundary for the confidence interval is estimated by

$$
\begin{cases}
z_{Q_1}^U = \max\left(z_{Q_1}^1, z_{Q_1}^2, \cdots, z_{Q_1}^{Nc}\right) \\
z_{V_3}^U = \max\left(z_{V_3}^1, z_{V_3}^2, \cdots, z_{V_3}^{Nc}\right) \\
z_{V_7}^U = \max\left(z_{V_7}^1, z_{V_7}^2, \cdots, z_{V_7}^{Nc}\right) \\
z_{V_{15}}^U = \max\left(z_{V_{15}}^1, z_{V_{15}}^2, \cdots, z_{V_{15}}^{Nc}\right)
\end{cases}
\tag{19}
$$

### 2.2.4.    Derivation of design flood hydrograph

In China, the design flood hydrograph for hydraulic structures is usually derived from the design flood events against a benchmark flood hydrograph, which is chosen from the observed flood processes (Ministry of Water Resources of People's Republic of China, 1996; Xiao et al., 2009, Yin et al., 2017). Suppose that a flood hydrograph consists of the features of annual maximum daily discharge (treated as flood peak), 3-day flood volume, 7-day flood volume and 15-day flood volume. The four features of the

benchmark flood hydrograph are denoted by $Q_1^B$, $V_3^B$, $V_7^B$ and $V_{15}^B$, respectively. The design flood hydrograph corresponding to the multivariate hydrologic design realization $\left(z_{Q_1}, z_{V_3}, z_{V_7}, z_{V_{15}}\right)$ can be derived by multiplying the benchmark flood hydrograph by different amplifiers, which are given as follows.

The amplifier $K_1$ for the annual maximum daily discharge is calculated by

$$
K_1 = \frac{z_{Q_1}}{Q_1^B}
\tag{20}
$$

The amplifier $K_{3-1}$ for the 3-day flood volume except for the annual maximum daily discharge is calculated by

$$
K_{3-1} = \frac{z_{V_3} - V\left(z_{Q_1}\right)}{V_3^T - V\left(Q_1^B\right)}
\tag{21}
$$



where $V(\cdot)$ is the operator transforming daily discharge into flood volume. The amplifier $K_{7-3}$ for the

7-day flood volume except for the 3-day flood volume is calculated by

$$K_{7-3} = \frac{z_{V_7} - z_{V_3}}{V_7^T - V_3^B} \tag{22}$$

Finally, the amplifier $K_{15-7}$ for the 15-day flood volume except for the 7-day flood volume is calculated

by

$$K_{15-7} = \frac{z_{V_{15}} - z_{V_7}}{V_{15}^T - V_7^B} \tag{23}$$

**3.   Study area and data set**

In this paper, the multivariate flood series from the Xijiang River, China (see Figure 2) are chosen to

perform the case study. As the western tributary as well as the main part of the Pearl River in South

China, the Xijiang River has a drainage area of 353,120 km$^2$ and flows a distance of 2,214 km. The

Xijiang River basin is located in the subtropical monsoon climate region, where the flood season lasts

from May to October. Owing to the humid climate and long flood season, river flooding has always

been a serious natural hazard in this basin.

With increasing urbanization during recent decades, more and more river training projects, such as

artificial levees, have been built in this basin to protect the urban region from river flooding. As a result,

more and more flood flow is constrained in the channel rather than overflowing, and thus the observed

flood of the river would increase (Xu et al., 2014). Another factor that significantly affects the flood

processes is reservoir regulation, especially after 2007, when two reservoirs with huge flood controlling

capacity were put into operation. One is named Longtan Reservoir, with a flood controlling capacity of

$5 \times 10^9$ m$^3$ and a catchment of 98,500 km$^2$, and the other is named Baise Reservoir, having a flood

controlling capacity of $1.64 \times 10^9$ m$^3$ and a catchment of 9,600 km$^2$.

The four-dimensional multivariate flood series, consisting of the annual maximum daily discharge

($Q_1$), annual maximum 3-day flood volume ($V_3$), annual maximum 7-day flood volume ($V_7$) and annual

maximum 15-day flood volume ($V_{15}$) observed at the Dahuangjiangkou gauge during the period from



1951 to 2012, are chosen as the study data. The Dahuangjiangkou gauge is located at the trunk stream of the Xijiang River, controlling a catchment of 294,669 km$^2$, about 83% of the total area of the Xijiang River basin.

In this study, both urbanization and reservoir regulation are considered as the potential driving force for the shifts of the flood processes. The effect of urbanization on the flood processes is quantified by the urban population ( *Pop* ). Given that the urban population data at the basin scale are not available and the vast majority of cities in the Xijiang River basin are distributed in Guangxi province, we use the urban population data of Guangxi province to represent those of the basin. The annual urban population data of Guangxi province during the observation period are obtained from the book of China Compendium of Statistics 1949–2008 (Department of Comprehensive Statistics of National Bureau of Statistics, 2010) and the website of the National Bureau of Statistics of PRC (http://www.stats.gov.cn/tjsj/ndsj/). The design life period for an assumed hydraulic structure is set to be from 2013 to 2100. The predicted urban population during the design life period is estimated based on the predicted growth rate of the urban population of China reported by He (2014). The reservoir index (*RI*), which depends on the catchment area and flood controlling capacity of reservoir, is used to quantify the effects of reservoir regulation on flood processes (López and Francés, 2013). Seen from Table 1, two reservoirs with flood controlling capacity have been completed during the observation period from 1951 to 2012, and two reservoirs are planned to be put into operation during the design life period. Figure 3 displays the evolution of the urban population and reservoir index during both the observation period and design life period.

## 4. Results

### 4.1. Nonstationary analysis for marginal distributions

The time-varying moments model is employed to perform the nonstationary analysis for each marginal distribution of the multivariate flood series $(Q_1, V_3, V_7, V_{15})$ from the Xijiang River. The urban population *Pop* and reservoir index RI are used as the candidate nonstationary indicators for the marginal distributions. The four candidate models of the time-varying margins are formulated as follows


$$\begin{aligned}
\mu_t &= \exp\left(\alpha_0 + \alpha_1 Pop_t + \alpha_2 RI_t\right) \\
\mu_t &= \exp\left(\alpha_0 + \alpha_1 Pop_t\right) \\
\mu_t &= \exp\left(\alpha_0 + \alpha_1 RI_t\right) \\
\mu_t &= \exp\left(\alpha_0\right)
\end{aligned}$$
(24)

In terms of the fitting quality assessed by AIC, the chosen model for each margin is displayed in Table 2, and also shown in Figure 4. Table 2 indicates that GEV distribution has the best fitting quality for the annual maximum daily discharge series $Q_1$, while Gamma distribution is chosen as the theoretical distribution for the flood volume series $V_3$, $V_7$ and $V_{15}$. The KS test indicates a satisfactory fitting effect for each marginal distribution. According to the regression functions of the location parameters $\mu$, the means of the flood series are generally positively related to the urban population $Pop$, while negatively related to the reservoir index $RI$. It can be concluded that the values of $\mu$ (referring to the means of the flood series) for all margins are nonstationary. This finding also reveals opposite roles played by urbanization and reservoir regulation on the flood processes of the Xijiang River. In particular, more artificial levees are required to protect urban region from flooding by constraining the flood flow in river channels due to increasing urbanization, which increases the flood flow observed in the river channel. The reservoir plays an active role in terms of flood control by reducing the flood discharge to downstream.

When it comes specifically to each margin of $\left(Q_1, V_3, V_7, V_{15}\right)$, the location parameters $\mu$ of the three short-duration flood series, i.e., $Q_1$, $V_3$ and $V_7$, are positively linked to $Pop$, while $RI$ is the driving factor reducing the location parameters $\mu$ of all flood series, including $Q_1$, $V_3$, $V_7$ and $V_{15}$. Owing to such difference in covariate selections, the short-duration flood series, including $Q_1$, $V_3$ and $V_7$, display asynchronous nonstationary behaviors with the long-duration flood series $V_{15}$ in the observation period of 1951~2012. As shown in Figure 4, $Q_1$, $V_3$ and $V_7$ present a significantly increasing trend during the period 1951~2005, especially since the 1980s, marking the beginning of a rapid-urbanization period in China. $V_{15}$ tends to follow a stationary process during the period 1951~2005. After 2006, when two reservoirs with the large flood controlling capacities were put into operation, all flood series, including



$Q_1$, $V_3$, $V_7$ and $V_{15}$, exhibit a sharp decline.

The predicted marginal distributions for $\left(Q_1, V_3, V_7, V_{15}\right)$ during the design life period from 2013 to 2100 are estimated through the time-varying moments model by replacing the observed covariates for $\mu$ with those predicted. Also illustrated in Figure 4, the mean values of $Q_1$, $V_3$ and $V_7$ during the design life period will increase with the growth of urban population, and then decrease sharply in 2023, when a larger reservoir named Datengxia is expected to be put into operation. After 2023, when no more reservoirs are planned, the predicted mean values of $Q_1$, $V_3$ and $V_7$ would be expected to reach their peaks in the mid-21th century followed by a slight declining trend because of shrinking urban population. Since $V_{15}$ is only related to $RI$, it would show an abrupt decline in 2023 due to the regulation of the Datengxia reservoir. In general, the predicted nonstationary marginal distributions for $Q_1$ and $V_3$ during the period of 2013~2100 roughly approximate to the marginal distributions under stationary assumption, while the predicted nonstationary marginal distributions for $V_7$ and $V_{15}$ exhibit smaller mean values than the stationary ones.

## 4.2.  Nonstationary dependence structure for $\left(Q_1, V_3, V_7, V_{15}\right)$

After estimating the nonstationary marginal distributions for $\left(Q_1, V_3, V_7, V_{15}\right)$, the multivariate dependence structure is constructed by the dynamic C-vine copula with $Q_1$ elected as the key variable. Similar to marginal distributions, the urban population $Pop$ and reservoir index RI are used as the candidate nonstationary indicators for the copula parameter. The four candidate models of time-varying dependence are formulated as follows

$$
\begin{aligned}
\theta_{c,t} &= \exp\left(\beta_0 + \beta_1 Pop_t + \beta_2 RI_t\right) \\
\theta_{c,t} &= \exp\left(\beta_0 + \beta_1 Pop_t\right) \\
\theta_{c,t} &= \exp\left(\beta_0 + \beta_1 RI_t\right) \\
\theta_{c,t} &= \exp\left(\beta_0\right)
\end{aligned}
\tag{25}
$$

Table 3 shows the estimation results of the dynamic C-vine copula. In terms of the fitting quality assessed by AIC, the bivariate pairs $\left(Q_1, V_3\right)$ and $\left(Q_1, V_7\right)$ exhibit the stationary dependence, while the copula parameter $\theta_{115}$ for pair $\left(Q_1, V_{15}\right)$ is found to be nonstationary and linked to both the urban





population *Pop* and reservoir index RI. It has been known that the margin of $Q_1$ displays the asynchronous nonstationarity behaviors with $V_{15}$ (see Table 2 and Figure 4). So the dependence nonstationarity of the pair $(Q_1, V_{15})$ could be possibly attributed to the asynchronous marginal nonstationarities. The goodness-of-fit examination for the nonstationary dependence structure of

$(Q_1, V_3, V_7, V_{15})$ suggests a satisfactory fitting effect by passing the PIT test at the 0.05 significance level.

According to the regression function, $\theta_{115}$ is negatively related to *Pop* while positively related to RI. In other words, the growing urbanization would weaken the multivariate flood dependence, while the reservoir regulation would play the opposite role and enhance the dependence. This finding indicates that human activities including urbanization and reservoir regulation not only change the

statistical characteristics of the individual flood series of $(Q_1, V_3, V_7, V_{15})$, but also affect the dependence of $(Q_1, V_3, V_7, V_{15})$.

Figure 5 illustrates the time variations of $\theta_{115}$ during the observation period of 1951~2012 as well as during the design life period of 2013~2100. It can be seen that $\theta_{115}$ has two significant upward change-points respectively occurring in 2007 and 2023 due to reservoir regulation. Besides the abrupt changes

described above, $\theta_{115}$ also exhibits an obvious decreasing trend with the growth of urban population from 1951 to the mid-21th century, followed by a slight increasing trend due to shrinking urban population. During the design life period, the predicted nonstationary $\theta_{115}$ suggests a weaker dependence for $(Q_1, V_3, V_7, V_{15})$ than the dependence under the stationary assumption, since it is usually smaller than the stationary estimation.

**4.3. Multivariate hydrologic design characterized by average annual reliability**

The multivariate hydrologic designs, characterized by the average annual reliability (AAR) respectively associated with OR, AND and Kendall exceedance probabilities, are estimated from the predicted nonstationary multivariate distribution for $(Q_1, V_3, V_7, V_{15})$ during the design life period from 2013 to 2100. Figures 6-9 (see the left columns) display the most-likely design events and the 90%



confidence intervals conditioned on the AAR varying from 0.01 to 0.99. It is found that the multivariate hydrologic design events associated with both the OR and Kendall exceedance probabilities exhibit the lower boundaries, while the design events associated with the AND exceedance probability display the upper boundaries. Since the copula density (such as Gumbel-Hougaard copula) is non-uniform and the flood variable generally follows the asymmetrical distribution (such as GEV and Gamma distributions),

the conditional probability density of the flood variable should be asymmetric and so should the confidence interval (Volpi and Fiori, 2012; Li et al., 2017). Seen from Figures 6-9, all confidence intervals are asymmetric.

    The design flood hydrographs are derived from the multivariate hydrologic designs against the benchmark flood hydrograph observed at 1988. Figure 10 illustrates the design flood hydrographs by

setting AAR equal to 0.90, 0.95 and 0.99. For any given multivariate flood event, the corresponding OR exceedance probability is larger than the AND exceedance probability, and the Kendall exceedance probability is somewhere in between (Vandenberghe et al., 2011). These differences among the OR, AND and Kendall exceedance probabilities induce the different design strategies. As displayed in Figure 10, the OR exceedance probability would generally yield the largest hydrologic design values,

followed by the Kendall and AND exceedance probabilities. It can be concluded that the design strategy associated with the OR exceedance probability is safer for hydraulic structures than those associated with the AND and Kendall exceedance probabilities, but on the other hand it might lead to an excessive cost for hydraulic structure construction. Oppositely, the hydrologic design associated with the AND exceedance probability has the smallest values, which could be more economical for the hydraulic

structure construction, but might bring the larger potential damage risk for hydraulic structures. The design strategy associated with the Kendall exceedance probability should be the moderate option between those associated with the AND and OR exceedance probabilities.

    To compare the design strategies under multivariate framework with those under univariate framework, we also calculate the univariate hydrologic design events from the predicted marginal

distributions. From Figures 6-9, it can be seen that the univariate hydrologic design events are exactly the lower boundaries of the multivariate hydrologic design events associated with the OR exceedance probability, as well as the upper boundaries of the design events associated with the AND exceedance



probability. Under the same given AAR, the hydrologic designs under the univariate framework are generally smaller than the most-likely design events associated with the OR exceedance probability, while larger than those associated with the AND exceedance probability, and most approximate to those associated with the Kendall exceedance probability. The comparisons of the flood hydrographs displayed in Figure 10 lead to the same findings.

### 4.4. Impacts of multivariate nonstationarity behaviors on hydrologic design values

In sections 4.1 and 4.2, both the marginal distribution and dependence structure of the multivariate flood distribution of $\left(Q_1, V_3, V_7, V_{15}\right)$ are found to be nonstationary. In order to illustrate how these nonstationarities act on the multivariate hydrologic designs, we estimate the multivariate hydrologic design events under the stationarity assumption, i.e., both marginal distributions and dependence structure are treated as stationary (see the right columns in Figures 6-9). Figure 4 suggests that both the predicted nonstationary marginal distributions for $Q_1$ and $V_3$ during the design life period are approximate to the stationary marginal distributions. Therefore, the nonstationary marginal distributions and the stationary marginal distributions yield the similar design values for $z_{Q_1}$ and $z_{V_3}$ (see Figures 6 and 7). The predicted nonstationary distributions for both $V_7$ and $V_{15}$ indicate the smaller mean values than the stationary distributions (see Figure 4), so the corresponding hydrologic design values estimated from the nonstationary marginal distributions are generally smaller than those estimated from the stationary marginal distributions (see Figures 8 and 9).

The nonstationary multivariate flood distribution during the design life period is predicted to exhibit a weaker dependence structure than the stationary distribution (see Figure 5). The dependence nonstationarity should have much subtler effect on the multivariate hydrologic design than the marginal nonstationarities (Xiong et al., 2015). For the purpose of displaying the individual role of dependence nonstationarity on multivariate hydrologic design, an artificial nonstationary condition for the multivariate flood distribution is set, that only the marginal nonstationarities are considered, while the dependence structure is treated as stationary. The results of the multivariate hydrologic design events are shown in the middle columns in Figures 6-9. In general, the dependence nonstationarity has less effect on the multivariate hydrologic designs compared the marginal nonstationarities, but some visible





differences in both the 90% confidence intervals and the most-likely design events can still be identified. The nonstationary and weaker dependence structure generally suggests a wider confidence intervals for the multivariate hydrologic design values.

## 5. Conclusion and discussion

Under nonstationary conditions, the statistical characteristics of both marginal distributions and
dependence structure of multivariate flood variables would vary with time. The multivariate flood distribution estimated from the historical information may fail to reflect the flood statistical characteristics in the future. As a result, the stationary-based hydrologic design could fail to deal with potential hydrologic risk of hydraulic structures. In engineering practice of hydrologic design, it is necessary for the designer to take account of the physical-related driving forces (such as human
activates and climate change) behind the nonstationarity of multivariate flood variables.

In this paper, we present the methods addressing the multivariate hydrologic design applying to the engineering practice under nonstationary conditions. First, a dynamic C-vine copula allowing for both time-varying marginal distributions and time-varying dependence structure is developed to capture the nonstationarities of multivariate flood distribution. Then, the multivariate hydrologic design under
nonstationary conditions is estimated through specifying the design criterion by average annual reliability. The most-likely design event and confidence interval are identified as the outcome of the multivariate hydrologic design. We choose multivariate flood series $\left(Q_1, V_3, V_7, V_{15}\right)$ from the Xijiang River, China, to perform the case study, and the main findings are given as follows.

For the multivariate flood series $\left(Q_1, V_3, V_7, V_{15}\right)$ of the Xijiang River, both urbanization and reservoir
regulation are driving force for the nonstationarities of the marginal distributions as well as of the dependence structure. The growth of urban population generally enlarges the mean value of the individual flood series, while weakens the dependence of $\left(Q_1, V_3, V_7, V_{15}\right)$. The increasing reservoir index has the opposite effects on the individual flood series as well as their dependence. Under the same given average annual reliability, the OR exceedance probability would yield the largest design values,
followed by the Kendall and AND exceedance probabilities. Nonstationarities in both marginal distributions and dependence structure can affect the outcome of the multivariate hydrologic design. For





the Xijiang River, the marginal nonstationarities play the dominant role affecting the multivariate hydrologic design.

For the practical implications of the hydrologic design methods developed in this paper, two
additional remarks are necessarily made.

The first one is about at least how long the observed flood data are required for multivariate and nonstationary hydrologic design. In theory, long enough observed flood data (or other extreme-value data) should be necessary to robustly estimate the flood distribution parameters and then yield the correct hydrologic design values. However, in reality, the length of data series is normally limited, thus
forcing us to use what we have at hands to do research or design works without fulfilling the theoretical assumptions or requirements. Some current literatures document that the univariate flood frequency analysis under stationary condition usually requires the flood data with a continuous period of at least 30 years (Ministry of Water Resources of People's Republic of China, 1996; Engeland et al., 2018; Kobierska et al., 2018). However, it is hard to find an orthodox answer to at least how long the observed
flood data should be required for flood frequency analysis under multivariate and/or nonstationary settings, since this issue has not yet been fully addressed by most previous studies. But one thing is for sure that, the multivariate and nonstationary hydrologic designs should naturally raise a higher request of data length, since nonstationary and multivariate models usually contain more parameters to be estimated.

The second remark is the tradeoff between reducing estimation bias and increasing estimators' variance. Nonstationary models generally improve the performance in fitting observation data by reducing estimation bias (Jiang et al., 2015b). But this improvement is usually achieved at the expense of increasing model complexity, such as adding more model parameters and introducing more nonstationary covariates, which might increase estimators' variance or induce additional sources of
model uncertainty (Serinaldi and Kilsby, 2015; Read and Vogel, 2016). Therefore, we should carefully balance the model fitting effect and the model complexity when employing multivariate and nonstationary hydrologic design in practice by keeping in mind the following two points: (1) the multivariate and nonstationary models should be not only kept effective but also kept as simple as possible to avoid over-fitting; and (2) to ensure a robust relationship of the distribution parameters to





the explanatory covariates, the chosen covariates should be physically-related to the flood processes and supported by a well-defined cause-effect analysis.

*Data availability*. All the data used in this study can be requested by contacting the corresponding
author C. Jiang at jiangcong@cug.edu.cn.

*Author contributions*. Cong Jiang and Lihua Xiong developed the main ideas. Cong Jiang and Lei Yan implemented the algorithms of the methods. Cong Jiang and Jianfan Dong collected the data used in the case study. Cong Jiang, Lihua Xiong and Chong-Yu Xu prepared the manuscript.

*Competing interests*. The authors declare that they have no conflict of interest.

*Acknowledgements*. This research is financially supported jointly by the Fundamental Research Funds for the Central Universities (Grant CUG170679), the National Natural Science Foundation of China (NSFC Grant 51525902), the Research Council of Norway (FRINATEK Project 274310), and the "111 Project" Fund of China (B18037), all of which are greatly appreciated.

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



**Table 1** Reservoirs information in the Xijiang River basin

| Reservoir | Catchment area (km$^2$) | Flood controlling capacity ($10^9$m$^3$) | Year put into operation |
|---|---|---|---|
| Longtan | 98,500 | 5.0 | 2006 |
| Baise | 9,600 | 1.64 | 2006 |
| Laokou | 72,368 | 0.36 | 2016 |
| Datengxia | 198,612 | 1.5 | 2023 (predicted) |






**Table 2** Results of nonstationary analysis for the marginal distributions of $\left( Q_1, V_3, V_7, V_{15} \right)$

| Flood variable | Distribution | $\mu$ | p_KS |
|----------------|--------------|-------|------|
| $Q_1$ | GEV | $\mu_t = e^{10.050+0.0212Pop_t-0.392RI_t}$ | 0.764 |
| $V_3$ | Gamma | $\mu_t = e^{1.866+0.0185Pop_t-0.379RI_t}$ | 0.897 |
| $V_7$ | Gamma | $\mu_t = e^{2.638+0.0119Pop_t-0.350RI_t}$ | 0.981 |
| $V_{15}$ | Gamma | $\mu_t = e^{3.285-0.276RI_t}$ | 0.975 |

p_KS stands for the p value of the KS test for marginal distributions.




**Table 3** Results of nonstationary analysis for the dependence structure of $(Q_1,V_3,V_7,V_{15})$

| Pair in C-vine copula | $\theta_c$ | p_PIT |
|---|---|---|
| $(Q_1,V_3)$ | 21.557 | 0.794 |
| $(Q_1,V_7)$ | 6.5778 | |
| $(Q_1,V_{15})$ | $\theta_{115,t}=1+e^{1.461-0.111Pop_t+1.706RI_t}$ | |
| $\left(F\left(u_{3,t}\mid u_{1,t}\right),F\left(u_{7,t}\mid u_{1,t}\right)\right)$ | 2.097 | |
| $\left(F\left(u_{3,t}\mid u_{1,t}\right),F\left(u_{15,t}\mid u_{1,t}\right)\right)$ | 1.236 | |
| $\left(F\left(u_{7,t}\mid u_{1,t},u_{3,t}\right),F\left(u_{15,t}\mid u_{1,t},u_{3,t}\right)\right)$ | 1.794 | |

p_PIT stands for p value of the PIT test for the C-vine copula.

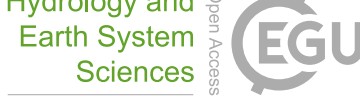



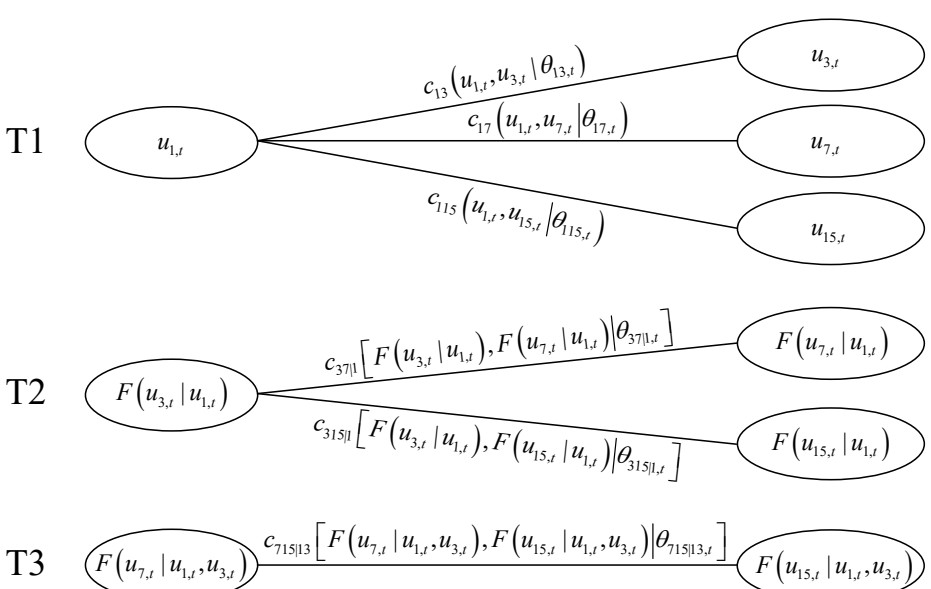

**Figure 1.** Decomposition of the four-dimensional C-vine copula.



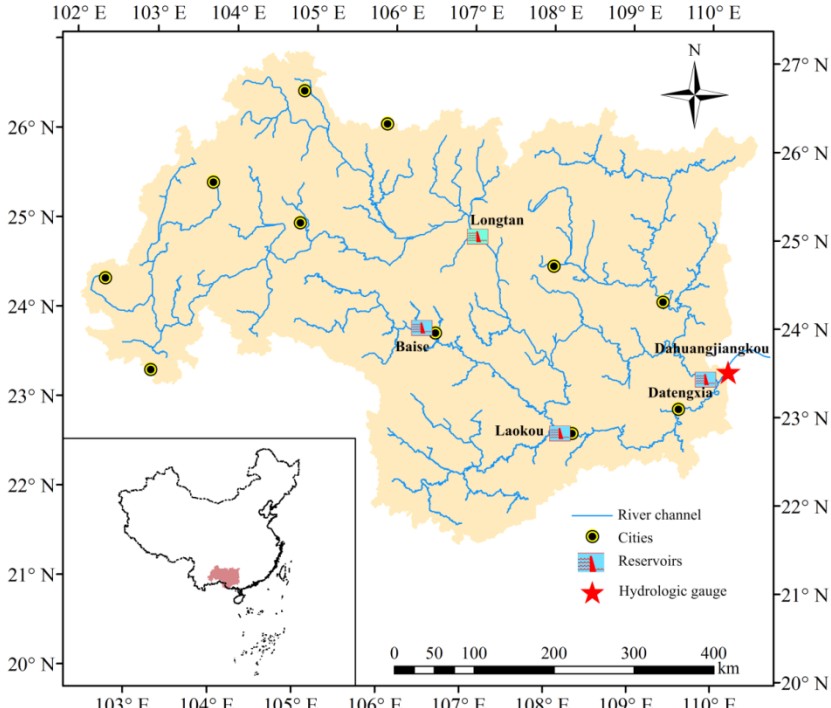

**Figure 2.** Map of the Xijiang River basin (above the Dahuangjiangkou gauge).





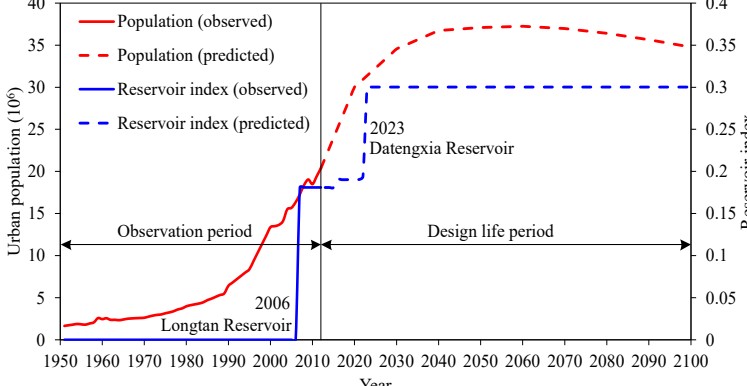

**Figure 3.** Evolutions of the urban population and reservoir index in both observation period and design life period.




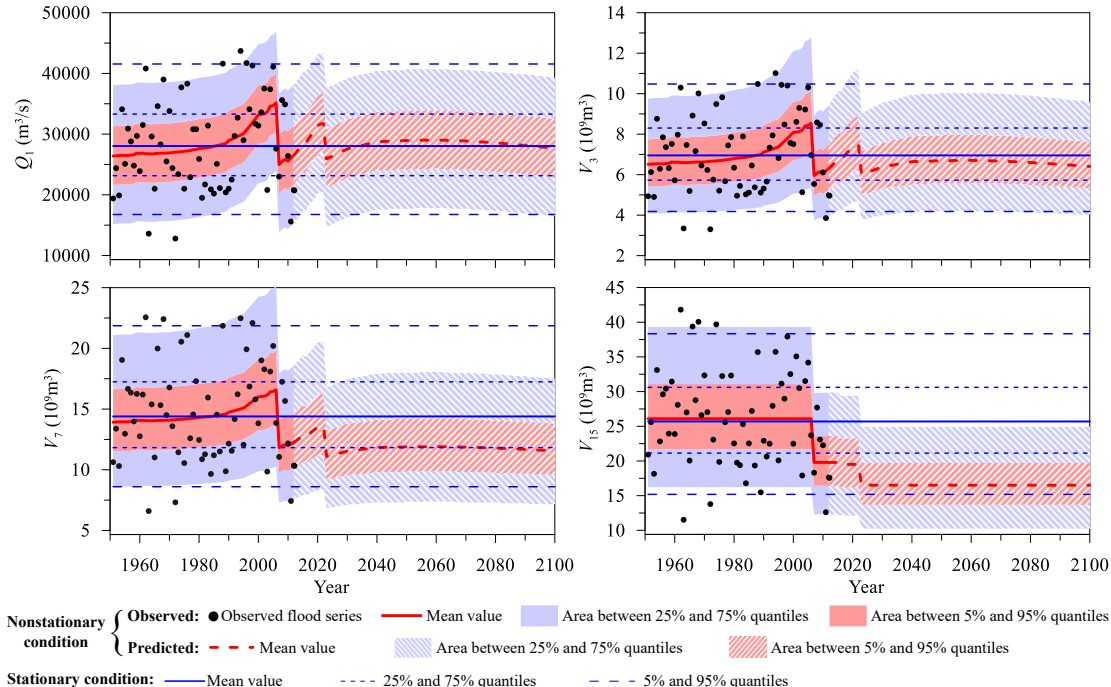

**Figure 4.** Nonstationary marginal distributions in both observation period and design life period.





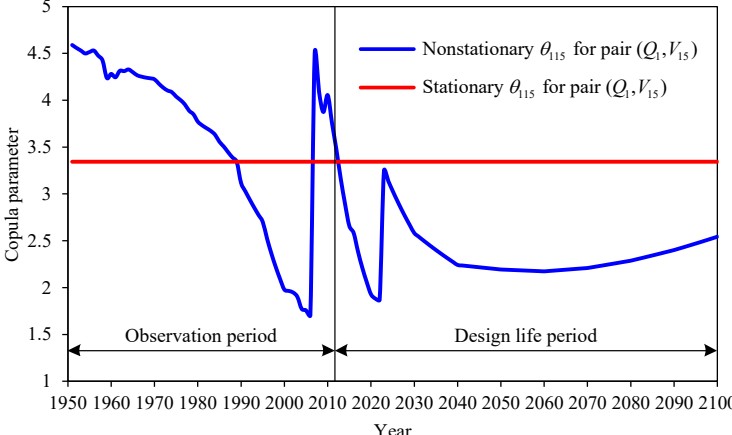


**Figure 5.** Nonstationary copula parameter for pair $\left(Q_1, V_{15}\right)$ in both observation period and design life period.







**Figure 6.** Design values of the annual maximum daily discharge for different AAR varying from 0.01 to 0.99 under three nonstationary conditions.





**Figure 7.** Design values of the 3-day flood volume for different AAR varying from 0.01 to 0.99 under three nonstationary conditions.






**Figure 8.** Design values of the 7-day flood volume for different AAR varying from 0.01 to 0.99 under three nonstationary conditions.





**Figure 9.** Design values of the 15-day flood volume for different AAR varying from 0.01 to 0.99 under three nonstationary conditions.








**Figure 10.** Design flood hydrographs associated with OR, AND and Kendall probabilities.