# Peer review of "Multivariate hydrologic design methods under nonstationary conditions and application to engineering practice"

_Hydrology and Earth System Sciences, 2018_

## Referee Comment (RC1) · Anonymous Referee #1 · 25 Jul 2018

**REVIEW REPORT**

**Journal:** Hydrology and Earth System Sciences
**Paper:** hess-2018-291
**Title:** Multivariate hydrologic design methods under nonstationary conditions and application to engineering practice
**Author(s):** Cong Jiang, Lihua Xiong, Lei Yan, Jianfan Dong and Chong-Yu Xu

**GENERAL COMMENTS.**

The paper deals with an issue of interest for the readers of HESS. In my opinion, a few critical issues must be fixed. Below, please find some indications: the objections should be read in a constructive way, since they may help the Authors improve the paper. Some useful bibliography is given at the end of this review.

**SPECIFIC COMMENTS.**

**Page(s) 3, Line(s) 48–49.**

    **Author(s).** . . . while the exceedance probability of a multivariate flood event could have multiple definitions (Salvadori et al., 2011; Vandenberghe et al., 2011).

    **Referee.** Here, an additional reference is Salvadori and De Michele (2004), where the non-uniqueness of return periods in a multivariate setting was first pointed out.

**Page(s) 3, Line(s) 54.**

    **Referee.** The citation to Salvadori and De Michele (2004) should be added here. Actually, there are (at least) four types of approaches, as recently outlined and discussed in Salvadori et al. (2016). The Authors are missing the Survival Kendall approach, first introduced in Salvadori et al. (2013): this latter avoids possible problems of divergence of the marginals on the boundaries of the domain.

**Page(s) 3, Line(s) 55.**

    **Author(s).** Due to changing climatic conditions, as well as some anthropogenic driving force. . .

    **Referee.** Here, the Authors should cite the paper by Milly et al. (2008).

**Page(s) 3, Line(s) 59–60.**

    **Author(s).** The multivariate flood distribution could exhibit more complex nonstationarity behaviors than the univariate distribution. . .

    **Referee.** Here, the Authors should cite the multivariate distributional approach outlined in Vezzoli et al. (2017)—where Sklar's Theorem representation is used to test nonstationarity—as well as the guidelines for multivariate change-point detection illustrated in Salvadori et al. (2018).

**Page(s) 4, Line(s) 87–88.**

    **Author(s).** These design criteria assess the risk or reliability of hydraulic structures. . .

    **Referee.** As a further approach involving Failure Probabilities, and design under nonstationarity and extreme value marginals, the Authors should cite the seminal paper by Salvadori et al. (2018).

**Page(s) 5, Line(s) 95–96.**

**Author(s).** ... these design events are generally not equivalent because their joint probability density values (i.e. likelihood) are usually different...

**Referee.** This fact was first pointed out in Salvadori et al. (2011).

**Page(s) 6, Line(s) 106–108.**

**Author(s).** ... the cases for higher-dimensional hydrologic designs as well as under nonstationary conditions have not yet been covered in current studies.

**Referee.** In Salvadori et al. (2018) appropriate confidence intervals are computed in a nonstationary case via suitable Monte Carlo techniques (the case study is a bivariate one, but the procedures can be generalized to any dimension).

**Page(s) 6–ff., Section(s) 2.1.**

In my opinion, the Authors should check for possible nonstationarity by using the functions provided by the R package "npcp" (Kojadinovic, 2017), which works both for the marginals and the copula, and provides approximate p-values concerning distributional changes.

**Page(s) 7, Eq(s) 1.**

The Authors should explain here the meaning of the undefined variable/parameter $t$ in Eq. (1). In addition, the mathematical notation used is wrong. Usually, in Probability, upper-case letters (e.g., $Q_{1,t}$) denote random variables, but here the arguments of the distribution functions $F$'s are real variables (i.e., lower-case letters). Please fix the notation throughout the manuscript.

**Page(s) 8, Line(s) 169.**

The covariates $(x_1, \ldots, x_k)$ should be specified here.

**Page(s) 8, Line(s) 172–174.**

**Author(s).** The higher-order distribution parameters such as scale and shape parameters are assumed to be kept constant to avoid possible larger uncertainty in parameter estimation, although they could also be nonstationary.

**Referee.** This assumption can make the work rather weak. In fact, here maxima are investigated, and the shape parameter plays a fundamental role in distributions like GEV or GPD, in order to rule the strength of the extremes: keeping these parameters constant could be unrealistic, and may yield strongly biased estimates. As shown in Salvadori et al. (2018), the uncertainties could really be large, but "constraining" the model by fixing the most relevant parameters is not a proper way to solve the problem. Please duly comment this point.

**Page(s) 8, Line(s) 178–180.**

**Author(s).** The goodness of fit (GOF) of the probability distributions is examined by using the Kolmogorov-Smirnov (KS) test (Frank and Massey, 1951).

**Referee.** This is a critical statistical point: how was the p-value computed? Just comparing the values of the KS test statistics with the ones reported in some table? This would be wrong. In fact, as is well known (e.g., simply read the help of Matlab), the KS test requires that the theoretical distribution be known a priori, it cannot be the fitted one. In the latter case, suitable (but simple) Monte Carlo techniques can be used to estimate an approximate p-value. Please fix the issue.

**Page(s) 9, Line(s) 191–192.**

**Author(s).** In this study the dependence structure of $(Q_1, V_3, V_7, V_{15})$ is constructed by the pair copula method.

**Referee.** To the best of my knowledge, the current software for vine-copulas does not provide reliable p-values of GoF tests: this may represent a statistical weakness of this approach. A comment is required here.

**Page(s) 11, Line(s) 227.**

The covariates $(x_1, \ldots, x_l)$ should be specified here.

**Page(s) 12, Line(s) 252–253.**

**Author(s).** ... the exceedance probability $p_t$ of $(q_1, v_3, v_7, v_{15})$...

**Referee.** What is the exceedance probability of a multivariate event? It should be properly defined here, or, at least, the Authors should put a reference to the next Section 2.2.2.

**Page(s) 12, Line(s) 262.**

**Author(s).** ... OR case, AND case and Kendall case...

**Referee.** Here, the best reference is Salvadori et al. (2016), where these cases are, for the first time, properly and rigorously defined in terms of suitable Hazard Scenarios based on the notions of Copulas and Lower/Upper Sets.

**Page(s) 13, Eq(s) 11.**

The $U$'s notation in Eq. (11) should be upper-case: the $U$'s used as arguments of the copula must be random variables, otherwise it make no sense to calculate a probability. Please fix this point.

**Page(s) 17, Line(s) 373–374.**

**Author(s).** The four candidate models of the time-varying margins are formulated as follows...

**Referee.** Please provide due comments/justifications about these choices.

**Page(s) 18, Line(s) 376–ff.**

**Author(s).** In terms of the fitting quality assessed by AIC...

**Referee.** The fitting procedure should work in the reverse sense. Viz., first the admissible distributions (among the ones of interest) are identified (if any) via a GoF test—e.g., KS, typically via a Monte Carlo algorithm, not by using values from statistical tables, as already mentioned above. Then, a "best" distribution is chosen (e.g., via AIC) only among the admissible ones. It makes no sense to compute the AIC of a non-admissible distribution. Please fix this point.

**Page(s) 18, Line(s) 380–382.**

**Author(s).** According to the regression functions of the location parameters $\mu$, the means of the flood series are generally positively related to the urban population *Pop*, while negatively related to the reservoir index *RI*.

**Referee.** This point is not clear. If a regression has been performed, the corresponding p-value should be shown, in order to decide whether the regression is statistically significant. In general, this work lacks of a solid statistical base. Please fix this point.

**Page(s) 19, Eq(s) 25.**

The formulas given in Eq.s (25) look different from the one shown in Eq. (7): is this correct?

**Page(s) 21, Line(s) 457–458.**

**Author(s).** These differences among the OR, AND and Kendall exceedance probabilities induce the different design strategies.

**Referee.** As discussed in Serinaldi (2015); Salvadori et al. (2016), comparing results induced by the usage of different Hazard Scenarios could be misleading, if not meaningless. The Hazard Scenario should be chosen a priori, not as a result of the consequences that the choice of a given scenario might entail. This looks like a methodological flaw. Please comment this point.

**Page(s) 23, Line(s) 511.**

**Author(s).** In this paper, we present the methods addressing the multivariate hydrologic design...

**Referee.** The claim "the methods" is incorrect, and too strong: you present "some possible methods". Please fix the sentence.

**References**

Kojadinovic, I., 2017. npcp: Some Nonparametric CUSUM Tests for Change-Point Detection in Possibly Multivariate Observations. R package version 0.1-9.
URL https://CRAN.R-project.org/package=npcp

Milly, P., Betancourt, J., Falkenmark, M., Hirsch, R., Kundzewicz, Z., Lettenmaier, D., Stouffer, R., 2008. Climate change - Stationarity is dead: Whither water management? Science 319 (5863), 573–574.

Salvadori, G., De Michele, C., 2004. Frequency analysis via copulas: theoretical aspects and applications to hydrological events. Water Resour. Res. 40, W12511, doi: 10.1029/2004WR003133.

Salvadori, G., De Michele, C., Durante, F., 2011. On the return period and design in a multivariate framework. Hydrol. Earth Syst. Sci. 15, 3293—3305.

Salvadori, G., Durante, F., De Michele, C., 2013. Multivariate return period calculation via survival functions. Water Resour. Res. 49, 2308–2311, doi: 10.1002/wrcr.20204.

Salvadori, G., Durante, F., De Michele, C., Bernardi, M., Petrella, L., 2016. A multivariate Copula-based framework for dealing with Hazard Scenarios and Failure Probabilities. Water Resources Research 52 (5), 3701–3721, doi: 10.1002/2015WR017225.

Salvadori, G., Durante, F., Michele, C. D., Bernardi, M., 2018. Hazard assessment under multivariate distributional change-points: Guidelines and a flood case study. Water 10 (6), 751–765.

Serinaldi, F., 2015. Dismissing return periods! Stochastic Environmental Research and Risk Assessment 29 (4), 1179–1189, doi: 10.1007/s00477-014-0916-1.

Vezzoli, R., Salvadori, G., De Michele, C., 2017. A distributional multivariate approach for assessing performance of climate-hydrology models. Scientific Reports 7:12071, doi: 10.1038/s41598-017-12343-1.
URL www.nature.com/scientificreports

---

## Referee Comment (RC2) · Anonymous Referee #2 · 6 Oct 2018

Dear editor Authors have presented the methods for applying the multivariate hydrologic design to the engineering practice under non-stationary conditions. Innovation of this paper is good but here there are main problems. 1- Materials and methods aren't described clearly. For accept of the manuscript, the test in mention section should be revised completely. 2- The reason for selection of C-vine as chosen method isn't specified. First mention a scientific for using of C-vine, then continue the rest of the manuscript. Because error may occur due to an incorrect selection of Vine.

---

## Referee Comment (RC3) · Anonymous Referee #3 · 11 Oct 2018

Review of "Multivariate hydrologic design methods under nonstationary conditions and application to engineering practice". Jiang et al. have developed a four-dimensional Vine copula for multivariate hydrologic designs under nonstationary conditions. Reading the abstract, I expected to read a throughout and well-organized study on such a hot topic. Going through the manuscript, I was a little bit disappointed, as the manuscript was not written in a coherent and clear way to reflect the concepts and methodology. There are serious concerns about the selection of different dimensions and also developing non-stationary models. Therefore, I cannot recommend the manuscript for publication in HESS journal in the current format. The manuscript needs substantial revision before considering for a potential publication. More details of my arguments

are provided below:

1- It is not clear why the authors select regulated flow time series for their study. Since the reservoir is above the gauge station, the flow time series is manipulated and does not represent the natural regime. Another question is that how do the authors address non-stationarity arising from global warming and land use change. And how do the authors separate the natural variability in flood series from the non-stationarity in their methodology.

2- Why do the authors select three flood volume dimensions, which are considered redundant? These variables are the same in nature and it is quite clear that the dependence between them should be high. The authors should explain why they do not select different variables representing different aspects of flow (with different nature) if they are really interested in applying a four-dimensional vine copula. Apparently, the whole process could be done using a bivariate copula. But if they are interested in developing a four-dimensional nonstationary-based vine copula, they should convince the readers why they select three of the dimensions from the same variable.

3- Why do the authors assume an exponential trend for the location parameters? How they make sure that there is not any other type of trend in the time series? What criteria is used to choose such an exponential trend? And why do not they use time dependent trend or any oscillation signal as covariates.

4- The reason that the authors do not assume any time dependent dependency in roots T2 and T3 is not clear.

5- Why do not the authors select the copula in eq 6 from the extreme copula families. And why do not they use any Goodness of Fit Test to select the best fitted copula from different copula families?

6- In equation 7, it is not clear that what is the covariate? Is time is the covariate? And again, why the authors use an exponential nonlinear trend to express the non-

stationarity in the copula permeameter? What if a linear or polynomial regression model is fitted well to express the trend in the copula parameters.

7- The authors talk about robustness of their model in lines 217 and 241. What is the definition of the robustness for these two cases?

8- The authors have not done any uncertainty analysis for estimation of the marginal and copula parameters through time.

9- Section 4.1 and 4.2 should move to the methodology, as they are not related to the results section.

10- Finally, the manuscript would greatly benefit from the input of a native English speaker.

---

## Author Comment (AC1) · 22 Oct 2018

**Rely to Referee #1**

**General comments**

The paper deals with an issue of interest for the readers of HESS. In my opinion, a few critical issues must be fixed. Below, please find some indications: the objections should be read in a constructive way, since they may help the Authors improve the paper. Some useful bibliography is given at the end of this review.

**Response:**

We are very grateful for your kind evaluation as well as insightful comments on this paper. All your comments and suggestions have been addressed in revising the manuscript.

**Specific comments**

**(1) Page(s) 3, Line(s) 48–49.**

**Author(s).** . . . while the exceedance probability of a multivariate flood event could have multiple definitions (Salvadori et al., 2011; Vandenberghe et al., 2011).

**Referee.** Here, an additional reference is Salvadori and De Michele (2004), where the non-uniqueness of return periods in a multivariate setting was first pointed out.

**Response:**

Thanks for this comment. The corresponding reference has been added in this sentence.

**Newly cited reference:**

Salvadori, G., and De Michele, C.: Frequency analysis via copulas: theoretical aspects and applications to hydrological events, Water Resour. Res., 40, W12511, https://doi.org/10.1029/2004WR003133, 2004.

**(2) Page(s) 3, Line(s) 54.**

**Referee.** The citation to Salvadori and De Michele (2004) should be added here. Actually, there are (at least) four types of approaches, as recently outlined and discussed in Salvadori et al. (2016). The Authors are missing the Survival Kendall approach, first introduced in Salvadori et al. (2013): this latter avoids possible problems of divergence of the marginals on the boundaries of the domain.

**Response:**

Thank you for this kind suggestion. The Survival Kendall approach and structural approach have been added in the review on the current multivariate design methods.

**Newly cited references:**

Salvadori, G., Durante, F., and De Michele, C.: Multivariate return period calculation via survival functions, Water Resour. Res., 49, 2308–2311, https://doi.org/10.1002/wrcr.20204, 2013.

Salvadori, G., Durante, F., De Michele, C., Bernardi, M., and Petrella, L.: A multivariate Copula-based framework for dealing with Hazard Scenarios and Failure Probabilities, Water Resour. Res., 52(5), 3701–3721, https://doi.org/10.1002/2015WR017225, 2016.

Requena, A. I., Mediero, L., and Garrote, L.: A bivariate return period based on copulas for hydrologic dam design: Accounting for reservoir routing in risk estimation, Hydrol. Earth Syst. Sci., 17(8), 3023–3038, https://doi.org/10.5194/hess-17-3023-2013, 2013.

Salvadori, G., Durante, F., Tomasicchio, G. R., and D'Alessandro, F. : Practical guidelines for the multivariate assessment of the structural risk in coastal and off-shore engineering, Coastal Eng., 95, 77–83, https://doi.org/10.1016/j.coastaleng.2014.09.007, 2015.

**(3) Page(s) 3, Line(s) 55.**

**Author(s).** Due to changing climatic conditions, as well as some anthropogenic driving force. . .

**Referee.** Here, the Authors should cite the paper by Milly et al. (2008).

**Response:**

Thank you for this suggestion. The paper by Milly et al. (2008) has been listed as the reference.

**Newly cited reference:**

Milly, P., Betancourt, J., Falkenmark, M., Hirsch, R., Kundzewicz, Z., Lettenmaier, D., and Stouffer, R.: Climate change - Stationarity is dead: Whither water management? Science, 319(5863), 573–574, https://doi.org/10.1126/science.1151915, 2008.

**(4) Page(s) 3, Line(s) 59–60.**

**Author(s).** The multivariate flood distribution could exhibit more complex nonstationarity behaviors than the univariate distribution. . .

**Referee.** Here, the Authors should cite the multivariate distributional approach outlined in Vezzoli et al. (2017)—where Sklar's Theorem representation is used to test nonstationarity—as well as the guidelines for multivariate change-point detection illustrated in Salvadori et al. (2018).

**Response:**

Thanks for your recommendation of these two papers, which are very helpful to support the view of this paper.

**Newly cited references:**

Vezzoli, R., Salvadori, G., and De Michele, C.: A distributional multivariate approach for assessing performance of climate-hydrology models, Scientific Reports, 7(12071), https://doi.org/10.1038/s41598-017-12343-1, 2017.

Salvadori, G., Durante, F., Michele, C. D., and Bernardi, M.: Hazard assessment under multivariate distributional change-points: Guidelines and a flood case study. Water 10 (6), 751–765, https://doi.org/10.3390/w10060751, 2018.

**(5) Page(s) 4, Line(s) 87–88.**

**Author(s).** These design criteria assess the risk or reliability of hydraulic structures. . .

**Referee.** As a further approach involving Failure Probabilities, and design under nonstationarity and extreme value marginals, the Authors should cite the seminal paper by Salvadori et al. (2018).

**Response:**

We have added this citation in the revised manuscript as follows:

Salvadori et al. (2018) associated hydrological designs with both given life times and failure probabilities to calculate the design values under nonstationarity.

**(6) Page(s) 5, Line(s) 95–96.**

**Author(s).** . . . these design events are generally not equivalent because their joint probability density values (i.e. likelihood) are usually different. . .

**Referee.** This fact was first pointed out in Salvadori et al. (2011).

**Response:**

We have added this citation in the revised manuscript.

**(7) Page(s) 6, Line(s) 106–108.**

**Author(s).** . . . the cases for higher-dimensional hydrologic designs as well as under nonstationary conditions have not yet been covered in current studies.

**Referee.** In Salvadori et al. (2018) appropriate confidence intervals are computed in a nonstationary case via suitable Monte Carlo techniques (the case study is a bivariate one, but the procedures can be generalized to any dimension).

**Response:**

In the revision, this sentence have been rephrased as follows:

Very few studies covered the cases for higher-dimensional hydrologic designs as well as under nonstationary conditions (Salvadori et al., 2018).

**(8) Page(s) 6–ff., Section(s) 2.1.**

In my opinion, the Authors should check for possible nonstationarity by using the functions provided by the R package "npcp" (Kojadinovic, 2017), which works both for the marginals and the copula, and provides approximate p-values concerning distributional changes.

**Response:**

Thank you for this constructive suggestion. In the revision work, we have employed the methods provide by R package "npcp" to detect the change points of the multivariate flood series. The results indicate that neither marginal distributions nor dependence (copula) display significant change points. But current finding shows that both the margins and dependence (copula parameter) are nonstationary due to the human activities of both urbanization and reservoir regulation. This difference in nonstationarity judgment should be attributed to the opposite roles of these two driving factors, that urbanization generally enlarges the mean value of the flood series and weakens the dependence, while reservoirs usually decrease the mean value and strengthen the dependence. In other words, the nonstationarities induced by these two factors could be offset by each other. As the result, the nonstationarities of the multivariate flood series might fail to be captured by the methods in "npcp". In the revised manuscript we have discussed this finding in the final of section 4.2 as follows:

Additionally, the change-point detection method based on Cramér-von Mises statistic (Bücher et al., 2014) is employed to detect the possible nonstationarities in both marginal distributions and dependence of the multivariate flood series $(Q_1, V_3, V_7, V_{15})$. For specific implementation steps of change-point detection, readers are referred to Bücher et al. (2014) and Kojadinovic (2017). The results indicate that neither marginal distributions nor dependence display change points at the 0.05 significance level, while the previous analysis suggests nonstationary margins and dependence due to the joint effects of urbanization and reservoir regulation. This difference in nonstationarity judgment should be attributed to the opposite roles of urbanization and reservoir regulation on shifting the multivariate flood distribution, that the former generally enlarges the mean values of the flood series and weakens their dependence, while the latter decreases the mean values and strengthen the dependence. In other words, the nonstationarities induced by these two factors could been offset by each other. As the result, the nonstationarities of $(Q_1, V_3, V_7, V_{15})$ might fail to be captured by the statistical method based on Cramér-von Mises statistic. This finding highlights the significance of cause-effect analysis in judging the nonstationarity of hydrological series (Serinaldi and Kilsby, 2015; Xiong et al., 2015).

**Newly cited references:**

Kojadinovic, I.: npcp: Some Nonparametric CUSUM Tests for Change-Point Detection in Possibly Multivariate Observations, R Package Version 0.1-9, Vienna, Austria, 2017.

Bücher, A., Kojadinovic, I., Rohmer, T., Segers, J.: Detecting changes in cross-sectional dependence in multivariate time series, J. Multivar. Anal., 132, 111–128, http://dx.doi.org/10.1016/j.jmva.2014.07.012, 2014.

**(9) Page(s) 7, Eq(s) 1.**

The Authors should explain here the meaning of the undefined variable/parameter $t$ in Eq. (1). In addition, the mathematical notation used is wrong. Usually, in Probability, upper-case letters (e.g., $Q1;t$) denote random variables, but here the arguments of the distribution functions $F$ 's are real variables (i.e., lower-case letters). Please fix the notation throughout the manuscript.

**Response:**

Thanks for comment. The symbol '$t$' denotes time measured by years, and this point has been explained in the revised manuscript. The mistakes in mathematical notation are also fixed throughout the manuscript.

**(10) Page(s) 8, Line(s) 169.**

The covariates $(x_1; : : : ; x_k)$ should be specified here.

**Response:**

The covariates $(x_1; : : : ; x_k)$ indicate the factors leading to marginal nonstationarities, i.e. urban population and reservoir index. In the revision, the covariates have been specified.

**(11) Page(s) 8, Line(s) 172–174.**

**Author(s).** The higher-order distribution parameters such as scale and shape parameters are assumed to be kept constant to avoid possible larger uncertainty in parameter estimation, although they could also be nonstationary.

**Referee.** This assumption can make the work rather weak. In fact, here maxima are investigated, and the shape parameter plays a fundamental role in distributions like GEV or GPD, in order to rule the strength of the extremes: keeping these parameters constant could be unrealistic, and may yield strongly biased estimates. As shown in Salvadori et al. (2018), the uncertainties could really be large, but "constraining" the model by fixing the most relevant parameters is not a proper way to solve the problem. Please duly comment this point.

**Response:**

Great thanks for this constructive comment. It is really true that the tradeoff between reducing estimation bias and increasing model uncertainty is debatable in the application of nonstationary models. In the revision, this sentence has been rephrased as follows:

In reality, all parameters of the flood distribution could be nonstationary, but in this paper only the nonstationarity of the location parameter $\mu$ (referring to the first moment or mean of flood series) is considered. Given the limited length of the flood series used in this study, the higher-order distribution parameters such as scale and shape parameters are fixed to avoid the possible large uncertainty.

In addition, a comment for this point is also added in the final paragraph of discussion section:

In this study, considering the limited length of the flood series, the nonstationarities in both higher-order marginal distribution parameters and copula parameters in roots T2 and T3 are ignored to avoid possible larger uncertainty in parameter estimation. It is also important to note that keeping these parameters constant could be unrealistic and yield biased estimates.

**(12) Page(s) 8, Line(s) 178–180.**
**Author(s).** The goodness of fit (GOF) of the probability distributions is examined by using the Kolmogorov-Smirnov (KS) test (Frank and Massey, 1951).
**Referee.** This is a critical statistical point: how was the *p*-value computed? Just comparing the values of the KS test statistics with the ones reported in some table? This would be wrong. In fact, as is well known (e.g., simply read the help of Matlab), the KS test requires that the theoretical distribution be known a priori, it cannot be the fitted one. In the latter case, suitable (but simple) Monte Carlo techniques can be used to estimate an approximate p-value. Please fix the issue.
**Response:**

Great thanks for this comment. In the revision, we have employed the Monte Carlo technique to estimate the *p*-value of the KS test for probability distributions. The results indicate the chosen distributions perform well in fitting the flood series.

**(13) Page(s) 9, Line(s) 191–192.**
**Author(s).** In this study the dependence structure of ($Q_1$; $V_3$; $V_7$; $V_{15}$) is constructed by the pair copula method.
**Referee.** To the best of my knowledge, the current software for vine-copulas does not provide reliable p-values of GoF tests: this may represent a statistical weakness of this approach. A comment is required here.
**Response:**

Thanks for this comment. It is really true that the GoF test methods for vine-copulas are very limited. In the study, the Probability Integral Transform (PIT) method is employed to perform the GoF test for vine copulas, and this method should be a reliable way to examine the GoF of vine copulas (Aas et al., 2009). In the revised manuscript, we have added a comment for this point and supplemented more information about the PIT test as follows:

The goodness of fit (GoF) tests for vine copulas are very limited, and the Probability Integral Transform (PIT) test (Rosenblatt, 1952) seems to be one of the few reliable methods (Aas et al.,

2009). Under the null hypothesis that the multivariate flood variables $\left(Q_1, V_3, V_7, V_{15}\right)$ follow a

given C-vine copula, the PIT converts the dependent flood variables into a new set of variables that are independent and uniformly distributed on $[0,1]^4$. Then the next step is to verify whether the resulting variables really are independent and uniform in $[0,1]$. This work can be finished by using chi-square test, and the significance level of the test is 0.05. For more details of the PIT test, reader are referred to Aas et al. (2009).

**(14) Page(s) 11, Line(s) 227.**
The covariates ($x_1$; $\because$; $x_l$) should be specified here.
**Response:**

The covariates have been specified in the revised manuscript.

**(15) Page(s) 12, Line(s) 252–253.**

**Author(s).** . . . the exceedance probability $p_t$ of $(q_1; v_3; v_7; v_{15})$. . .

**Referee.** What is the exceedance probability of a multivariate event? It should be properly defined here, or, at least, the Authors should put a reference to the next Section 2.2.2.

**Response:**

The exceedance probability $p_t$ of $(q_1; v_3; v_7; v_{15})$ indicates the danger of flood event, and is defined as the occurrence probability of the event more dangerous than $(q_1; v_3; v_7; v_{15})$ in a specific hazard scenario. In the revision, we have added this definition.

**(16) Page(s) 12, Line(s) 262.**

**Author(s).** . . . OR case, AND case and Kendall case. . .

**Referee.** Here, the best reference is Salvadori et al. (2016), where these cases are, for the first time, properly and rigorously defined in terms of suitable Hazard Scenarios based on the notions of Copulas and Lower/Upper Sets.

**Response:**

In the revision, we have added this reference.

**(17) Page(s) 13, Eq(s) 11.**

The $U$'s notation in Eq. (11) should be upper-case: the $U$'s used as arguments of the copula must be random variables, otherwise it make no sense to calculate a probability. Please fix this point.

**Response:**

Great thanks. We have made the correction.

**(18) Page(s) 17, Line(s) 373–374.**

**Author(s).** The four candidate models of the time-varying margins are formulated as follows. . .

**Referee.** Please provide due comments/justifications about these choices.

**Response:**

Thank you for this constructive suggestion. In the revised manuscript, we have added some comments for these four candidate models.

As above, the first equation defines the most complex nonstationary model, where it is assumed that both urban population and reservoir index are the driving factors of marginal distributions; the second and third equations indicate that the marginal nonstationarity is only linked to urban population and reservoir index, respectively; and the fourth and final one is the simplest and stationary model.

**(19) Page(s) 18, Line(s) 376–ff.**

**Author(s).** In terms of the fitting quality assessed by AIC. . .

**Referee.** The fitting procedure should work in the reverse sense. Viz., first the admissible distributions (among the ones of interest) are identified (if any) via a GoF test—e.g., KS, typically via a Monte Carlo algorithm, not by using values from statistical tables, as already mentioned above. Then, a "best" distribution is chosen (e.g., via AIC) only among the admissible ones. It makes no sense to compute the AIC of a non-admissible distribution. Please fix this point.

**Response:**

We have adjusted the fitting procedure for marginal distributions in the revision according to this comment.

**(20) Page(s) 18, Line(s) 380–382.**

**Author(s).** According to the regression functions of the location parameters $\mu$, the means of the flood series are generally positively related to the urban population *Pop* , while negatively related to the reservoir index *RI*.

**Referee.** This point is not clear. If a regression has been performed, the corresponding p-value should be shown, in order to decide whether the regression is statistically significant. In general, this work lacks of a solid statistical base. Please fix this point.

**Response:**

In the revision, the $p$-values of regression parameters have been calculated and displayed in Table 2. The results indicate that all parameters are statistically significant at the 0.05 level.

**(21) Page(s) 19, Eq(s) 25.**

The formulas given in Eq.s (25) look different from the one shown in Eq. (7): is this correct?

**Response:**

Thanks. It is a typing mistake. In the revision, we have fixed it.

**(22) Page(s) 21, Line(s) 457–458.**

**Author(s).** These differences among the OR, AND and Kendall exceedance probabilities induce the different design strategies.

**Referee.** As discussed in Serinaldi (2015); Salvadori et al. (2016), comparing results induced by the usage of different Hazard Scenarios could be misleading, if not meaningless. The Hazard Scenario should be chosen a priori, not as a result of the consequences that the choice of a given scenario might entail. This looks like a methodological flaw. Please comment this point.

**Response:**

Great thanks for this insightful comment. It is really true that the choice of design strategy is generally priori, and depends on the specific situations of design requirements and mechanisms of failure for hydraulic structures. In the revision, we have added the following comment:

These differences among the OR, AND and Kendall exceedance probabilities indicate the different design strategies. In engineering practice, the choice of design strategy is generally priori, and depends on the specific situations of design requirements and mechanisms of failure for hydraulic structures (Serinaldi, 2015; Salvadori et al., 2016).

**Newly cited reference:**

Serinaldi, F.: Dismissing return periods!, Stoch. Eev. Res. Risk. A., 29 (4), 1179–1189, http://dx.doi.org/10.1007/s00477-014-0916-1, 2015.

**(23) Page(s) 23, Line(s) 511.**

**Author(s).** In this paper, we present the methods addressing the multivariate hydrologic design. . .

**Referee.** The claim "the methods" is incorrect, and too strong: you present "some possible methods". Please fix the sentence.

**Response:**

In the revision, we have rephrased this sentence according your suggestion.

---

## Author Comment (AC2) · 22 Oct 2018

**Rely to Referee #2**

Authors have presented the methods for applying the multivariate hydrologic design to the engineering practice under non-stationary conditions. Innovation of this paper is good but here there are main problems.

**Response:**

Great thanks for your positive evaluation as well as professional comments on our study. All your comments have been addressed in revising the manuscript.

(1) Materials and methods aren't described clearly. For accept of the manuscript, the test in mention section should be revised completely.

**Response:**

In the revision, we have added an Appendix to explain the methods used this study in more detail. The tests for the methods (including GoF tests for both marginal distributions and copula) have been revised completely and supplemented more explanations. The revised manuscript should be clear for readers.

The revised tests for the methods are given as follows:

The goodness of fit (GoF) of the probability distributions is examined by using the Kolmogorov-Smirnov (KS) test (Frank and Massey, 1951). The p-value of the KS test is simulated from Monte Carlo method, and the significance level of the test is 0.05.

The goodness of fit (GoF) tests for vine copulas are very limited, and the Probability Integral Transform (PIT) test (Rosenblatt, 1952) seems to be one of the few reliable methods (Aas et al., 2009). Under the null hypothesis that the multivariate flood variables $\left(Q_1, V_3, V_7, V_{15}\right)$ follow a given C-vine copula, the PIT converts the dependent flood variables into a new set of variables that are independent and uniformly distributed on $[0,1]^4$. Then the next step is to verify whether the resulting variables really are independent and uniform in $[0,1]$. This work can be finished by using chi-square test, and the significance level is set to be 0.05. For more details of the PIT test, readers are referred to Aas et al. (2009).

(2) The reason for selection of C-vine as chosen method isn't specified. First mention a scientific for using of C-vine, then continue the rest of the manuscript. Because error may occur due to an incorrect selection of Vine.

**Response:**

Thanks for this comment. In this study, the annual maximum daily discharge ($Q_1$), annual maximum 3-day flood volume ($V_3$), annual maximum 7-day flood volume ($V_7$) and annual maximum 15-day flood volume ($V_{15}$) are chosen to define the multivariate flood series. It is known that flood peak (e.g., $Q_1$) is the dominant feature quantifying flood event as well as the key factor in hydrologic design (Ministry of Water Resources of People's Republic of China, 1996). The C-vine is more suitable when there is a key variable governing multivariate dependence, therefore it is reasonable to employ C-vine copula to construct the joint distribution of ($Q_1$, $V_3$, $V_7$, $V_{15}$) with $Q_1$ elected as the key variable. In the revision, we have specified this explanation in the second paragraph of section 3.1.2.

---

## Author Comment (AC5) · 22 Oct 2018

**Reply to referee # 3**

**General comment**

Jiang et al. have developed a four-dimensional Vine copula for multivariate hydrologic designs under nonstationary conditions. Reading the abstract, I expected to read a throughout and well-organized study on such a hot topic. Going through the manuscript, I was a little bit disappointed, as the manuscript was not written in a coherent and clear way to reflect the concepts and methodology. There are serious concerns about the selection of different dimensions and also developing non-stationary models. Therefore, I cannot recommend the manuscript for publication in HESS journal in the current format. The manuscript needs substantial revision before considering for a potential publication.

**Response:**

We are very grateful for your constructive comments on this manuscript. All your comments are very valuable improving this manuscript, and have been carefully considered in the revision. Please see our point-to-point replay below.

**Specific comments**

1- It is not clear why the authors select regulated flow time series for their study. Since the reservoir is above the gauge station, the flow time series is manipulated and does not represent the natural regime. Another question is that how do the authors address non-stationarity arising from global warming and land use change. And how do the authors separate the natural variability in flood series from the non-stationarity in their methodology.

**Response:**

Great thanks for this kind comment. For the purpose of water resources development and hydropower generation, it is hard to find a river which is not impacted by reservoirs, especially in the rapidly developing China. The natural flow processes of the rivers in China as well as many places over the world have been significantly regulated by reservoir operation, which has be a significant force inducing nonstationarities of flood series. Therefore, reservoir operation should be seriously concerned in the downstream flood risk analysis and hydrologic design (López and Francés, 2013; Xiong et al., 2015). In this study, we select the Xijiang River located in Southwest China, where large numbers reservoirs have been built, to perform the case study and explore the effect of reservoir regulation on the multivariate flood distribution. In the revised manuscript, this explanation has been added in the third paragraph of section 2.

Global warming can indeed lead to flood nonstationarity by altering the climatic conditions of the basin. As for the study area concerned in this paper, the climatic conditions dominating flood processes, such as extreme precipitation, seem to be stationary over the past decades (Yang et al., 2010). That is the reason why we do not consider the effect of global warming on the flood distribution. Some previous studies have proved that the land use change such as urbanization is an important factor leading to nonstationarity of the flood series of the Xijiang River. In this study, the effect of urbanization has been concerned by introducing urban population of the basin as an explanatory covariate of the nonstationarity of the flood series. Based on cause-effect analysis, the nonstationarity of the flood series is attributed to the joint effect of reservoir regulation and urbanization, so the natural variability in flood series is not considered in this case study. In the revised manuscript, we have added the above explanations in the final of the third paragraph of

section 2.

**Newly cited reference:**

Yang, T., Shao, Q., Hao, Z., Chen, Xi., Zhang, Z., Xu, C.-Y., Sun, L.: Regional frequency analysis and spatio-temporal pattern characterization of rainfall extremes in the Pearl River Basin, China, J. Hydrol., 380, 386–405, https://doi.org/10.1016/j.jhydrol.2009.11.013, 2010.

2- Why do the authors select three flood volume dimensions, which are considered redundant? These variables are the same in nature and it is quite clear that the dependence between them should be high. The authors should explain why they do not select different variables representing different aspects of flow (with different nature) if they are really interested in applying a four-dimensional vine copula. Apparently, the whole process could be done using a bivariate copula. But if they are interested in developing a four-dimensional nonstationary-based vine copula, they should convince the readers why they select three of the dimensions from the same variable.

**Response:**

Thanks for this comment. In this paper, the flood series from the Xijiang River of China are chosen as the study data to illustrate the multivariate design methods under nonstationary conditions. According to the regulation for calculating design flood of China (Ministry of Water Resources of People's Republic of China, 1996), deriving flood hydrographs for hydraulic structures requires not only flood peak but also flood volumes with different durations, such as 3 days, 7 days, 15 days and 30 days (Xiao et al., 2009; Xiong et al., 2015; Li et al., 2017). Since this paper selects the hydrological design in China to perform the case study, the multivariate flood series should consist of flood peak and flood volume variables with different durations. It is necessary to note that the proposed methods can be extended to other multivariate flood series, such as consisting of flood peak, flood volume and flood duration, which represent different aspects of flow.

In the revision, the above explanation has been added in the second paragraph of section 'Study area and data set' and the first paragraph of section 'methods'. The structure of the manuscript is also reorganized. The section 'Study area and data set' (i.e. section 3 in the original manuscript) is listed as section 2, and the section 'methods' (i.e. section 2 in the original manuscript) is moved to section 3. Thus, the reasons why we select three flood volume dimensions have been explained before illustrating the methods. This adjustment would make it clear and logical for readers to understand the methods of this paper.

3- Why do the authors assume an exponential trend for the location parameters? How they make sure that there is not any other type of trend in the time series? What criteria is used to choose such an exponential trend? And why do not they use time dependent trend or any oscillation signal as covariates.

**Response:**

Thanks for this comment. In this paper, the domain of the location parameters (referring to the first moment or mean of flood series) should be $(0, +\infty)$. The exponential function enables the location parameters to always satisfy the domain and be meaningful, and this is something that some other functions such as linear and polynomial model are incapable of doing. In quite a few studies, the exponential function is selected to describe the nonstationarity of flow series (Vogel et al., 2011; Jiang et al., 2015; Read and Vogel, 2016; Yan et al., 2017). In the revised manuscript, the above reason has been supplemented in the second paragraph of section 3.1.1. In addition, we also have

employed a linear function to build the relationship between location parameters and explanatory covariates. The results suggest that the difference between the linear model and exponential model is very tiny. That is to say, the trend of the location parameter is mainly determined by the variations of the explanatory covariates, rather than the functions for describing the relationship between location parameter and covariates.

It is known that the flood series of this paper are impacted by both reservoir operation and urbanization. Therefore, reservoir index and urban population are definitely the more meaningful covariates than time variable in terms of mechanism of the flood nonstationarities. This is the reason why we use reservoir index and urban population as the explanatory covariates to describe the nonstationarity of location parameters rather than the time dependent trend. In the observation period, the climatic conditions (such as extreme precipitation) of the study basin do not exhibit significant time variation, and therefore the oscillation signals indicating the climatic nonstationary are not considered in this study. In the revised manuscript, the above explanations have been added in the second paragraph of section 3.1.1.

4- The reason that the authors do not assume any time dependent dependency in roots T2 and T3 is not clear.

**Response:**

Thanks for this comment. It is true that the dependency (quantified by copula parameters) in roots T2 and T3 could be time dependent or nonstationary, at least theoretically. However, the length of the observed flood series is quite limited, and the estimation of the parameters in roots T2 and T3 depends on the estimated parameters in T1. It means that the estimation of the parameters in roots T2 and T3 contains more sources of uncertainty, and a complex nonstationary model would probably lead to far greater uncertainty than the simple stationary model. Therefore, we prefer to a stationary dependency in roots T2 and T3. In the revision, we have added the reasons stated above. In addition, this point is also stressed in the final paragraph of discussion section.

5- Why do not the authors select the copula in equation 6 from the extreme copula families. And why do not they use any Goodness of Fit Test to select the best fitted copula from different copula families?

**Response:**

In this study, we employ the Gumbel-Hougaard copula (as expressed by equation 6), which is an extreme-value copula and widely used in hydrology field, to construct the dynamic C-vine copula. In flood frequency analysis, the upper tail of flood distribution deserves more attention because it allows to quantify the risks of the flood events with greater dangerous. It is known that the Gumbel-Hougaard copula accounts for the upper tail dependence, and therefore is well-suited to the dependence structure of multivariate flood distribution (Salvadori et al., 2007; Zhang and Singh, 2007; Xiong et al., 2015). This is the reason why the Gumbel-Hougaard copula is selected in this study, and has been added in the fifth paragraph of section 3.1.2. In addition, we also have considered more copulas used in hydrology filed, such as Frank, Clayton and $t$ student copulas. The results indicate that Gumbel-Hougaard copula has the best fitting quality in terms of AIC.

6- In equation 7, it is not clear that what is the covariate? Is time is the covariate? And again, why the authors use an exponential nonlinear trend to express the non-stationarity in the copula

parameter? What if a linear or polynomial regression model is fitted well to express the trend in the copula parameters.

**Response:**

The covariate in equation 7 denotes the explanatory variable describing the nonstationarity of copula parameter. Based on the initial cause-effect analysis for the nonstationarities of the flood series, the covariates used in this study contain reservoir index and urban population, both of which are the factors leading to the variation of flood processes. In the revision, the covariates in equation 7 have been specified as reservoir index and urban population. Since time variable has no cause-effect relationship with the nonstationarity of flood series, it is not used as the covariate of copula parameter.

In this study, we select Gumbel-Hougaard copula to construct the joint distribution of multivariate flood series, and the domain of the copula parameter is $[1, +\infty)$. To satisfy the domain range of the copula parameter under any conditions, an exponential model expressed as

$$\theta_{c,t} = 1 + \exp(\cdot)$$

is employed to build the relationship between copula parameter and covariates. A linear or polynomial regression model could induce the copula parameter beyond the domain and to be meaningless. In the revision, the above explanation has been added in the fifth paragraph of section 3.1.2.

7- The authors talk about robustness of their model in lines 217 and 241. What is the definition of the robustness for these two cases?

**Response:**

The so-called robustness suggests the uncertainty of the estimated parameter. In the revision we have specified this explanation.

8- The authors have not done any uncertainty analysis for estimation of the marginal and copula parameters through time.

**Response:**

Thanks for this constructive comment. In the revision, we have performed an uncertainty analysis for the estimated marginal and copula parameters using the parametric bootstrap method. The 95% uncertainty intervals of the estimated parameters are displayed in Table 2 and Table 3.

9- Section 4.1 and 4.2 should move to the methodology, as they are not related to the results section.

**Response:**

In the revision, the context about methodology in section 4.1 and 4.2 has been moved to the methods section.

10- Finally, the manuscript would greatly benefit from the input of a native English speaker.

**Response:**

The revised manuscript will be proofed by a native English speaker before it is uploaded.

---

## Author Response (AR1)

**Dear Editor,**

On behalf of my co-authors, I would like to express our sincere thanks to you and the anonymous reviewers for the efforts on reviewing our manuscript entitled "Multivariate hydrologic design methods under nonstationary conditions and application to engineering practice" (ID: hess-2018-291).

In accordance with the comments of the reviewers, this manuscript has been revised. All the comments raised by the reviewers are very professional and have been addressed in the revision of the manuscript. A point-by-point response to the comments and the relevant changes made in the manuscript are presented as appendix to this letter. Attached please find the revised manuscript with all revisions marked in red color. We hope the revision of the manuscript will meet with the approval of the reviewers and editors for publication in HESS.

Great thanks to you and the three anonymous reviewers once again for the efforts on the substantial improvement of our paper.

With all best wishes

Yours

Cong Jiang
On behalf of my co-authors

**Reply to Referee #1**

**General comments**

The paper deals with an issue of interest for the readers of HESS. In my opinion, a few critical issues must be fixed. Below, please find some indications: the objections should be read in a constructive way, since they may help the Authors improve the paper. Some useful bibliography is given at the end of this review.

**Response:**

We are very grateful for your kind evaluation as well as insightful comments on this paper. All your comments and suggestions have been addressed in revising the manuscript.

**Specific comments**

**(1) Page(s) 3, Line(s) 48–49.**

**Author(s).** . . . while the exceedance probability of a multivariate flood event could have multiple definitions (Salvadori et al., 2011; Vandenberghe et al., 2011).

**Referee.** Here, an additional reference is Salvadori and De Michele (2004), where the non-uniqueness of return periods in a multivariate setting was first pointed out.

**Response:**

Thanks for this comment. The corresponding reference has been added in this sentence.

**Newly cited reference:**

Salvadori, G., and De Michele, C.: Frequency analysis via copulas: theoretical aspects and applications to hydrological events, Water Resour. Res., 40, W12511, https://doi.org/10.1029/2004WR003133, 2004.

**(2) Page(s) 3, Line(s) 54.**

**Referee.** The citation to Salvadori and De Michele (2004) should be added here. Actually, there are (at least) four types of approaches, as recently outlined and discussed in Salvadori et al. (2016). The Authors are missing the Survival Kendall approach, first introduced in Salvadori et al. (2013): this latter avoids possible problems of divergence of the marginals on the boundaries of the domain.

**Response:**

Thank you for this kind suggestion. The Survival Kendall approach and structural approach have been added in the review on the current multivariate design methods.

**Newly cited references:**

Salvadori, G., Durante, F., and De Michele, C.: Multivariate return period calculation via survival functions, Water Resour. Res., 49, 2308–2311, https://doi.org/10.1002/wrcr.20204, 2013.

Salvadori, G., Durante, F., De Michele, C., Bernardi, M., and Petrella, L.: A multivariate Copula-based framework for dealing with Hazard Scenarios and Failure Probabilities, Water Resour. Res., 52(5), 3701–3721, https://doi.org/10.1002/2015WR017225, 2016.

Requena, A. I., Mediero, L., and Garrote, L.: A bivariate return period based on copulas for hydrologic dam design: Accounting for reservoir routing in risk estimation, Hydrol. Earth Syst. Sci., 17(8), 3023–3038, https://doi.org/10.5194/hess-17-3023-2013, 2013.

Salvadori, G., Durante, F., Tomasicchio, G. R., and D'Alessandro, F. : Practical guidelines for

the multivariate assessment of the structural risk in coastal and off-shore engineering, Coastal Eng., 95, 77–83, https://doi.org/10.1016/j.coastaleng.2014.09.007, 2015.

**(3) Page(s) 3, Line(s) 55.**
**Author(s).** Due to changing climatic conditions, as well as some anthropogenic driving force...
**Referee.** Here, the Authors should cite the paper by Milly et al. (2008).
**Response:**

Thank you for this suggestion. The paper by Milly et al. (2008) has been listed as the reference.
**Newly cited reference:**

Milly, P., Betancourt, J., Falkenmark, M., Hirsch, R., Kundzewicz, Z., Lettenmaier, D., and Stouffer, R.: Climate change - Stationarity is dead: Whither water management? Science, 319(5863), 573–574, https://doi.org/10.1126/science.1151915, 2008.

**(4) Page(s) 3, Line(s) 59–60.**
**Author(s).** The multivariate flood distribution could exhibit more complex nonstationarity behaviors than the univariate distribution...
**Referee.** Here, the Authors should cite the multivariate distributional approach outlined in Vezzoli et al. (2017)—where Sklar's Theorem representation is used to test nonstationarity—as well as the guidelines for multivariate change-point detection illustrated in Salvadori et al. (2018).
**Response:**

Thanks for your recommendation of these two papers, which are very helpful to support the view of this paper.
**Newly cited references:**

Vezzoli, R., Salvadori, G., and De Michele, C.: A distributional multivariate approach for assessing performance of climate-hydrology models, Scientific Reports, 7(12071), https://doi.org/10.1038/s41598-017-12343-1, 2017.

Salvadori, G., Durante, F., Michele, C. D., and Bernardi, M.: Hazard assessment under multivariate distributional change-points: Guidelines and a flood case study. Water 10 (6), 751–765, https://doi.org/10.3390/w10060751, 2018.

**(5) Page(s) 4, Line(s) 87–88.**
**Author(s).** These design criteria assess the risk or reliability of hydraulic structures...
**Referee.** As a further approach involving Failure Probabilities, and design under nonstationarity and extreme value marginals, the Authors should cite the seminal paper by Salvadori et al. (2018).
**Response:**

We have added this citation in the revised manuscript as follows:

Salvadori et al. (2018) associated hydrologic designs with both given life times and failure probabilities to calculate bivariate design values under nonstationarity.

**(6) Page(s) 5, Line(s) 95–96.**
**Author(s).** ... these design events are generally not equivalent because their joint probability

density values (i.e. likelihood) are usually different. . .

**Referee.** This fact was first pointed out in Salvadori et al. (2011).

**Response:**

We have added this citation in the revised manuscript.

**(7) Page(s) 6, Line(s) 106–108.**

**Author(s).** . . . the cases for higher-dimensional hydrologic designs as well as under nonstationary conditions have not yet been covered in current studies.

**Referee.** In Salvadori et al. (2018) appropriate confidence intervals are computed in a nonstationary case via suitable Monte Carlo techniques (the case study is a bivariate one, but the procedures can be generalized to any dimension).

**Response:**

In the revision, this sentence have been rephrased as follows:

The most-likely design event and confidence interval for the bivariate hydrologic design under stationary conditions have been identified (Salvadori et al., 2011; Volpi and Fiori, 2012; Li et al., 2017; Yin et al., 2017; Salvadori et al., 2018); however, very few studies have focused on hydrologic designs with higher dimensions under nonstationary conditions.

**(8) Page(s) 6–ff., Section(s) 2.1.**

In my opinion, the Authors should check for possible nonstationarity by using the functions provided by the R package "npcp" (Kojadinovic, 2017), which works both for the marginals and the copula, and provides approximate p-values concerning distributional changes.

**Response:**

Thank you for this constructive suggestion. In the revision work, we have employed the methods provide by R package "npcp" to detect the change points of the multivariate flood series. The results indicated that neither the marginal distributions nor dependence displayed change points at the 0.05 significance level, whereas the previous analysis suggested nonstationary margins and dependence due to the joint effects of urbanization and reservoir regulation. These aforementioned inconsistencies could be attributed to the opposite roles of urbanization and reservoir regulation on shifting of the multivariate flood distribution, with urbanization generally enlarging the mean values of the flood series and weakening their dependence, and reservoir regulation decreasing the mean values and strengthening the dependence. In other words, the nonstationarities induced by these two factors may have offset each other. As a result, the nonstationarities of the flood series might have not been captured by the "npcp" method. In the revised manuscript we have discussed this finding in the final of section 4.2 as follows:

In addition, the change-point detection method based on the Cramér-von Mises statistic (Bücher et al., 2014) was employed to detect possible nonstationarities in both the marginal distributions and dependence of the multivariate flood series $(Q_1, V_3, V_7, V_{15})$ . Readers are referred to Bücher et al. (2014) and Kojadinovic (2017) for specific steps to implement change-point detection. The results indicated that neither the marginal distributions nor dependence displayed change points at the 0.05 significance level, whereas the previous analysis suggested nonstationary margins and dependence due to the joint effects of urbanization and reservoir

regulation. These aforementioned inconsistencies could be attributed to the opposite roles of urbanization and reservoir regulation on shifting of the multivariate flood distribution, with urbanization generally enlarging the mean values of the flood series and weakening their dependence, and reservoir regulation decreasing the mean values and strengthening the dependence. In other words, the nonstationarities induced by these two factors may have offset each other. As a result, the nonstationarities of $(Q_1, V_3, V_7, V_{15})$ might have not been captured by the statistical method based on the Cramér-von Mises statistic. This finding highlights the significance of cause-effect analysis in judging the nonstationarity of hydrologic series (Serinaldi and Kilsby, 2015; Xiong et al., 2015).

**Newly cited references:**

Kojadinovic, I.: npcp: Some Nonparametric CUSUM Tests for Change-Point Detection in Possibly Multivariate Observations, R Package Version 0.1-9, Vienna, Austria, 2017.

Bücher, A., Kojadinovic, I., Rohmer, T., Segers, J.: Detecting changes in cross-sectional dependence in multivariate time series, J. Multivar. Anal., 132, 111–128, http://dx.doi.org/10.1016/j.jmva.2014.07.012, 2014.

**(9) Page(s) 7, Eq(s) 1.**
The Authors should explain here the meaning of the undefined variable/parameter *t* in Eq. (1). In addition, the mathematical notation used is wrong. Usually, in Probability, upper-case letters (e.g., *Q*1*;t*) denote random variables, but here the arguments of the distribution functions *F* 's are real variables (i.e., lower-case letters). Please fix the notation throughout the manuscript.

**Response:**

  Thanks for comment. The symbol '*t*' denotes time measured by years, and this point has been explained in the revised manuscript. The mistakes in mathematical notation are also fixed throughout the manuscript.

**(10) Page(s) 8, Line(s) 169.**
The covariates ($x_1; : : : ; x_k$) should be specified here.

**Response:**

  The covariates ($x_1; : : : ; x_k$) indicate the factors leading to marginal nonstationarities, i.e. urban population (*Pop*) and reservoir index (*RI*). In the revision, the covariates have been specified as *Pop* and *RI*.

**(11) Page(s) 8, Line(s) 172–174.**
**Author(s).** The higher-order distribution parameters such as scale and shape parameters are assumed to be kept constant to avoid possible larger uncertainty in parameter estimation, although they could also be nonstationary.

**Referee.** This assumption can make the work rather weak. In fact, here maxima are investigated, and the shape parameter plays a fundamental role in distributions like GEV or GPD, in order to rule the strength of the extremes: keeping these parameters constant could be unrealistic, and may yield strongly biased estimates. As shown in Salvadori et al. (2018), the uncertainties could really be large, but "constraining" the model by fixing the most relevant parameters is not a proper way to solve the problem. Please duly comment this point.

**Response:**

Great thanks for this constructive comment. It is really true that the tradeoff between reducing estimation bias and increasing model uncertainty is debatable in the application of nonstationary models. In the revision, this sentence has been rephrased as follows:

In reality, all parameters of the flood distribution can be nonstationary, but this paper only considered nonstationarity of the location parameter $\mu$ (referring to the first moment or mean of the flood series). Given the limited length of the flood series used in this study, the higher-order distribution parameters such as scale and shape parameters were fixed to avoid possible large uncertainty.

In addition, a comment for this point is also added in the final paragraph of discussion section:

Nonstationary models generally improve performance in fitting observation data by reducing estimation bias (Jiang et al., 2015b), but this is usually achieved at the expense of increasing model complexity, such as adding more model parameters and introducing more nonstationary covariates, which might induce additional sources of model uncertainty (Serinaldi and Kilsby, 2015; Read and Vogel, 2016). The nonstationarities in the present study for both higher-order marginal distribution parameters and copula parameters in roots T2 and T3 were ignored to avoid possible larger uncertainty in parameter estimation considering the limited length of the flood series. It is also important to note that keeping these parameters constant could be unrealistic and could result in biased estimates.

**(12) Page(s) 8, Line(s) 178–180.**
**Author(s).** The goodness of fit (GOF) of the probability distributions is examined by using the Kolmogorov-Smirnov (KS) test (Frank and Massey, 1951).
**Referee.** This is a critical statistical point: how was the *p*-value computed? Just comparing the values of the KS test statistics with the ones reported in some table? This would be wrong. In fact, as is well known (e.g., simply read the help of Matlab), the KS test requires that the theoretical distribution be known a priori, it cannot be the fitted one. In the latter case, suitable (but simple) Monte Carlo techniques can be used to estimate an approximate p-value. Please fix the issue.
**Response:**

Great thanks for this comment. In the revision, we have employed the Monte Carlo technique to estimate the *p*-value of the KS test for probability distributions. The results indicate the chosen distributions perform well in fitting the flood series.

**(13) Page(s) 9, Line(s) 191–192.**
**Author(s).** In this study the dependence structure of ($Q_1$; $V_3$; $V_7$; $V_{15}$) is constructed by the pair copula method.
**Referee.** To the best of my knowledge, the current software for vine-copulas does not provide reliable p-values of GoF tests: this may represent a statistical weakness of this approach. A comment is required here.
**Response:**

Thanks for this comment. It is really true that the GoF test methods for vine-copulas are

very limited. In the study, the GoF test for vine copulas is performed using the Probability Integral Transform (PIT) method, which should be a reliable way (Aas et al., 2009). In the revised manuscript, we have added a comment for this point and supplemented more information about the PIT test as follows:

The available GoF tests for vine copulas are very limited, with the Probability Integral Transform (PIT) test (Rosenblatt, 1952) appearing to be reliable (Aas et al., 2009). Under a null hypothesis of the multivariate flood variables $(Q_1, V_3, V_7, V_{15})$ following a given C-vine copula, the PIT converts the dependent flood variables into a new set of variables that are independent and uniformly distributed on $[0,1]^4$. The GoF of vine copulas can be obtained through determining whether the resulting variables are independent and uniform in $[0,1]$. For more details of the PIT test, readers are referred to Aas et al. (2009).

**(14) Page(s) 11, Line(s) 227.**
The covariates $(x_1, \ldots, x_l)$ should be specified here.
**Response:**
The covariates have been specified as urban population and reservoir index in the revised manuscript.

**(15) Page(s) 12, Line(s) 252–253.**
**Author(s).** . . . the exceedance probability $p_t$ of $(q_1; v_3; v_7; v_{15})$. . .
**Referee.** What is the exceedance probability of a multivariate event? It should be properly defined here, or, at least, the Authors should put a reference to the next Section 2.2.2.
**Response:**
The exceedance probability $p_t$ of $(q_1; v_3; v_7; v_{15})$ indicates the danger of flood event, and is defined as the occurrence probability of the event more dangerous than $(q_1; v_3; v_7; v_{15})$ in a specific hazard scenario. In the revision, we have added this definition.

**(16) Page(s) 12, Line(s) 262.**
**Author(s).** . . . OR case, AND case and Kendall case. . .
**Referee.** Here, the best reference is Salvadori et al. (2016), where these cases are, for the first time, properly and rigorously defined in terms of suitable Hazard Scenarios based on the notions of Copulas and Lower/Upper Sets.
**Response:**
In the revision, we have added this reference.

**(17) Page(s) 13, Eq(s) 11.**
The $U$'s notation in Eq. (11) should be upper-case: the $U$'s used as arguments of the copula must be random variables, otherwise it make no sense to calculate a probability. Please fix this point.
**Response:**
Great thanks. We have made the correction.

**(18) Page(s) 17, Line(s) 373–374.**

**Author(s).** The four candidate models of the time-varying margins are formulated as follows. . .
**Referee.** Please provide due comments/justifications about these choices.
**Response:**

Thank you for this constructive suggestion. In the revised manuscript, we have added some comments for these four candidate models following Eq. (3).

As above, the first equation defines the most complex nonstationary model where it is assumed that both *RI* and *Pop* are the driving factors of marginal distributions; the second and third equations illustrate that the marginal nonstationarity is linked only to *RI* and *Pop*, respectively and; the final equations represent the simplest model and stationary model.

**(19) Page(s) 18, Line(s) 376–ff.**
**Author(s).** In terms of the fitting quality assessed by AIC. . .
**Referee.** The fitting procedure should work in the reverse sense. Viz., first the admissible distributions (among the ones of interest) are identified (if any) via a GoF test—e.g., KS, typically via a Monte Carlo algorithm, not by using values from statistical tables, as already mentioned above. Then, a "best" distribution is chosen (e.g., via AIC) only among the admissible ones. It makes no sense to compute the AIC of a non-admissible distribution. Please fix this point.
**Response:**

We have adjusted the fitting procedure for marginal distributions in the revision according to this comment.

**(20) Page(s) 18, Line(s) 380–382.**
**Author(s).** According to the regression functions of the location parameters $\mu$, the means of the flood series are generally positively related to the urban population *Pop* , while negatively related to the reservoir index *RI*.
**Referee.** This point is not clear. If a regression has been performed, the corresponding p-value should be shown, in order to decide whether the regression is statistically significant. In general, this work lacks of a solid statistical base. Please fix this point.
**Response:**

In the revision, the *p*-values of regression parameters have been calculated and displayed in Table 2. The results indicate that all parameters are statistically significant at the 0.05 level.

**(21) Page(s) 19, Eq(s) 25.**
The formulas given in Eq.s (25) look different from the one shown in Eq. (7): is this correct?
**Response:**

Thanks. It is a typing mistake and has been fixed. Please see Eq. (7) in the revised manuscript.

**(22) Page(s) 21, Line(s) 457–458.**
**Author(s).** These differences among the OR, AND and Kendall exceedance probabilities induce the different design strategies.
**Referee.** As discussed in Serinaldi (2015); Salvadori et al. (2016), comparing results induced by the usage of different Hazard Scenarios could be misleading, if not meaningless. The Hazard

Scenario should be chosen a priori, not as a result of the consequences that the choice of a given scenario might entail. This looks like a methodological flaw. Please comment this point.
**Response:**

Great thanks for this insightful comment. It is really true that the choice of design strategy is generally priori, and is dependent on the specific design requirements and mechanisms of failure for hydraulic structures. In the revision, we have added the following comment:

These differences among the OR, AND and Kendall exceedance probabilities indicate the different design strategies. It must be noted that the choice of design strategy in engineering practice is usually priori and is dependent on the specific design requirements and mechanisms of failure for hydraulic structures (Serinaldi, 2015; Salvadori et al., 2016).

**Newly cited reference:**

Serinaldi, F.: Dismissing return periods!, Stoch. Eev. Res. Risk. A., 29 (4), 1179–1189, http://dx.doi.org/10.1007/s00477-014-0916-1, 2015.

**(23) Page(s) 23, Line(s) 511.**

**Author(s).** In this paper, we present the methods addressing the multivariate hydrologic design. . .

**Referee.** The claim "the methods" is incorrect, and too strong: you present "some possible methods". Please fix the sentence.

**Response:**

In the revision, we have rephrased this sentence as below:

The present study introduced possible methods for addressing multivariate hydrologic design for application in engineering practice under nonstationary conditions.

Authors have presented the methods for applying the multivariate hydrologic design to the engineering practice under non-stationary conditions. Innovation of this paper is good but here there are main problems.

**Response:**

Great thanks for your positive evaluation as well as professional comments on our study. All your comments have been addressed in revising the manuscript.

(1) Materials and methods aren't described clearly. For accept of the manuscript, the test in mention section should be revised completely.

**Response:**

In the revision, we have added an Appendix to explain the methods used this study in more detail. The tests for the methods (including GoF tests for both marginal distributions and copula) have been revised completely and supplemented more explanations. The revised manuscript should be clear for readers.

Appendix to explain the methods is given as follows:

**Appendix**

**1. Calculating multivariate exceedance probabilities**

**1.1 OR exceedance probability** (formulated by Eq. (9) in the paper)

Since the multivariate cumulative function $F\left(q_1, v_3, v_7, v_{15} | \mathbf{\theta}_t\right)$ has no analytical expression, the OR exceedance probability $p_t^{or}$ at time $t$ is calculated by the Monte Carlo method as follows:

(1) Calculate the marginal probabilities $\left(u_1, u_3, u_7, u_{15}\right)$ of $\left(q_1, v_3, v_7, v_{15}\right)$;

(2) Generate $m$ samples $\left(u_{1,i}, u_{3,i}, u_{7,i}, u_{15,i}\right)$ $\left(i = 1, 2, ..., m\right)$ from the C-vine copula;

(3) Calculate $F\left(q_1, v_3, v_7, v_{15} | \mathbf{\theta}_t\right) = \dfrac{1}{m+1} \sum_{i=1}^{m} \mathbf{1}\left(u_{1,i} \leq u_1, u_{3,i} \leq u_3, u_{7,i} \leq u_7, u_{15,i} \leq u_{15}\right)$;

(4) Calculate $p_t^{or} = 1 - F\left(q_1, v_3, v_7, v_{15} | \mathbf{\theta}_t\right)$.

**1.2 AND exceedance probability** (formulated by Eq. (10) in the paper)

The AND exceedance probability $p_t^{and}$ at time $t$ is calculated by the Monte Carlo method as follows:

(1) Calculate the marginal probabilities $\left(u_1, u_3, u_7, u_{15}\right)$ of $\left(q_1, v_3, v_7, v_{15}\right)$;

(2) Generate $m$ samples $\left(u_{1,i}, u_{3,i}, u_{7,i}, u_{15,i}\right)$ $\left(i = 1, 2, ..., m\right)$ from the C-vine copula;

(3) Calculate $p_t^{and} = \dfrac{1}{m+1} \sum_{i=1}^{m} \mathbf{1}\left(u_{1,i} \geq u_1, u_{3,i} \geq u_3, u_{7,i} \geq u_7, u_{15,i} \geq u_{15}\right)$.

**1.3 The Kendall exceedance probability** (formulated by Eq. (11) and Eq. (12) in the paper)

The Kendall exceedance probability $p_t^{ken}$ at time $t$ is calculated by the Monte Carlo method as follows:

(1) Calculate the marginal probabilities $(u_1, u_3, u_7, u_{15})$ of $(q_1, v_3, v_7, v_{15})$;

(2) Calculate $\rho_t = F(q_1, v_3, v_7, v_{15} | \boldsymbol{\theta}_t)$ (see calculation steps 2–3 for OR exceedance probability);

(3) Generate $m$ samples $(u_{1,i}, u_{3,i}, u_{7,i}, u_{15,i})$ $(i = 1, 2, ..., m)$ from the C-vine copula;

(4) For $j = 1, 2, ..., m$, calculate $v_j = \dfrac{1}{m+1} \sum_{i=1}^{m} \mathbf{1}(u_{1,i} \le u_{1,j}, u_{3,i} \le u_{3,j}, u_{7,i} \le u_{7,j}, u_{15,i} \le u_{15,j})$;

(5) Calculate $K_C(\rho_t) = \dfrac{1}{m} \sum_{i=1}^{m} \mathbf{1}(v_i \le \rho_t)$;

(6) Calculate $p_t^{ken} = 1 - K_C(\rho_t)$.

**2. Generating the multivariate design event sample** (formulated by Eq. (17) in the paper)

To calculate the most-likely design event and confidence interval conditioned on $AAR = \eta$, we need to generate the numerous multivariate design event samples $(z_{Q_1}, z_{V_3}, z_{V_7}, z_{V_{15}})$ by using the Monte Carlo method. Here, we give the details of generating the design event samples as follows:

(1) Define the total number of design event samples $N$ and the initial number of the design event sample $i = 0$;

(2) Generate a random integer (denoted by $t_r$) among ($T_1$, $T_1$+1,…, $T_2$);

(3) Generate a random sample $(z_{Q_1}, z_{V_3}, z_{V_7}, z_{V_{15}})$ following the multivariate distribution $F(z_{Q_1}, z_{V_3}, z_{V_7}, z_{V_{15}} | \boldsymbol{\theta}_{t_r})$ with the distribution parameter vector $\boldsymbol{\theta}_{t_r}$;

(4) Calculate the annual exceedance probability for each year throughout the period from $T_1$ to $T_2$;

(5) Calculate $AAR$ during the period from $T_1$ to $T_2$;

(6) If $|AAR - \eta| < \varepsilon$ ( where is a very small value, such as 0.0001) , $i = i+1$;

(7) If $i < N$, repeat steps (2)–(6).

The tests for the methods are revised as follows:

The goodness of fit (GoF) of the probability distributions was examined by the Kolmogorov-Smirnov (KS) test with significance level set to 0.05 (Frank and Massey, 1951). The p-value of the KS test was simulated using the Monte Carlo method.

The available GoF tests for vine copulas are very limited, with the Probability Integral Transform (PIT) test (Rosenblatt, 1952) appearing to be reliable (Aas et al., 2009). Under a null hypothesis of the multivariate flood variables $(Q_1, V_3, V_7, V_{15})$ following a given C-vine copula,

the PIT converts the dependent flood variables into a new set of variables that are independent and uniformly distributed on $[0,1]^4$. The GoF of vine copulas can be obtained through determining whether the resulting variables are independent and uniform in $[0,1]$. For more details of the PIT test, readers are referred to Aas et al. (2009).

(2) The reason for selection of C-vine as chosen method isn't specified. First mention a scientific for using of C-vine, then continue the rest of the manuscript. Because error may occur due to an incorrect selection of Vine.

**Response:**

Thanks for this comment. In this study, the annual maximum daily discharge ($Q_1$), annual maximum 3-day flood volume ($V_3$), annual maximum 7-day flood volume ($V_7$) and annual maximum 15-day flood volume ($V_{15}$) are chosen to define the multivariate flood series. It is known that flood peak (e.g., $Q_1$) is the dominant feature quantifying flood event as well as the key factor in hydrologic design (Ministry of Water Resources of People's Republic of China, 1996). The C-vine is more suitable when there is a key variable governing multivariate dependence, therefore it is reasonable to employ C-vine copula to construct the joint distribution of ($Q_1,V_3,V_7,V_{15}$) with $Q_1$ elected as the key variable. In the revision, we have specified this explanation in the second paragraph of section 3.1.2 as follows:

It is known that flood peak (e.g., $Q_1$) is the dominant feature quantifying a flood event as well as the key factor in hydrologic design (Ministry of Water Resources of People's Republic of China, 1996). The C-vine is more suitable when there is a key variable governing multivariate dependence (Aas et al., 2009). In this case, the C-vine was employed to construct the joint distribution of ($Q_1,V_3,V_7,V_{15}$) with $Q_1$ elected as the key variable.

**Reply to referee # 3**

**General comment**

Jiang et al. have developed a four-dimensional Vine copula for multivariate hydrologic designs under nonstationary conditions. Reading the abstract, I expected to read a throughout and well-organized study on such a hot topic. Going through the manuscript, I was a little bit disappointed, as the manuscript was not written in a coherent and clear way to reflect the concepts and methodology. There are serious concerns about the selection of different dimensions and also developing non-stationary models. Therefore, I cannot recommend the manuscript for publication in HESS journal in the current format. The manuscript needs substantial revision before considering for a potential publication.

**Response:**

We are very grateful for your constructive comments on this manuscript. All your comments are very valuable improving this manuscript, and have been carefully considered in the revision.

**Specific comments**

1- It is not clear why the authors select regulated flow time series for their study. Since the reservoir is above the gauge station, the flow time series is manipulated and does not represent the natural regime. Another question is that how do the authors address non-stationarity arising from global warming and land use change. And how do the authors separate the natural variability in flood series from the non-stationarity in their methodology.

**Response:**

Great thanks for this kind comment. For the purpose of flood control and hydropower generation, it is hard to find a river which is not impacted by reservoirs, especially in the rapidly developing China. The natural flow processes of the rivers in China as well as many regions over the world have been significantly regulated by reservoir operation, which has be a significant force inducing nonstationarities of flood series. Therefore, reservoir operation should be seriously concerned in the downstream flood risk analysis and hydrologic design (López and Francés, 2013; Xiong et al., 2015). In this study, we select the Xijiang River located in South China, where several huge reservoirs have been built, to perform the case study and explore the effect of reservoir regulation on the multivariate flood distribution.

Global warming will likely result in flood nonstationarity by altering the climatic conditions of the basin. As for the study area concerned in this paper, the climatic conditions dominating flood processes, such as extreme precipitation, seem to be stationary over the past decades (Yang et al., 2010). That is the reason why we do not consider the effect of global warming on the flood distribution. Some previous studies have proved that the land use change such as urbanization is an important factor leading to nonstationarity of the flood series of the Xijiang River. In this study, the effect of urbanization has been concerned by introducing urban population of the basin as an explanatory covariate of the nonstationarity of the flood series. Based on cause-effect analysis, the nonstationarity of the flood series is mainly attributed to the joint effect of reservoir regulation and urbanization, so the natural variability in flood series is not considered in this case study.

In the revised manuscript, we have added the above explanations in the third paragraph of section 2 as follows:

Rapid urbanization over recent decades has resulted in increasing river regulation projects built in the XRB, such as artificial levees for protecting urban areas from river flooding. As a result, flood flow has become increasingly constrained to the channel rather than overflow to the floodplain, resulting in an increase in the observed river flood flow (Xu et al., 2014). For the purpose of flood control and hydropower generation, it is hard to find a river which is not impacted by reservoirs, particularly in a rapidly developing China. Reservoir regulation has become an increasingly significant factor affecting flood processes of the XRB, and should be seriously considered within downstream flood risk analysis and hydrologic design, particularly since 2007 when two reservoirs with considerable flood control capacities were put into operation. These are the Longtan and Baise reservoirs with flood control capacities of $5 \times 10^9 \, \text{m}^3$ and $1.64 \times 10^9 \, \text{m}^3$ and catchment areas of 98,500 km$^2$ and 9,600 km$^2$, respectively. Climate change will likely result in flood nonstationarity by altering climatic conditions of the basin. Climatic conditions dominating flood processes in the XRB, such as extreme precipitation, appear to have been stationary over the past decades (Yang et al., 2010). Therefore, the current study introduced only urbanization and reservoir regulation as potential driving forces of nonstationarity of the flood series, and ignored the effect of climate change.

**Newly cited reference:**

Yang, T., Shao, Q., Hao, Z., Chen, Xi., Zhang, Z., Xu, C.-Y., Sun, L.: Regional frequency analysis and spatio-temporal pattern characterization of rainfall extremes in the Pearl River Basin, China, J. Hydrol., 380, 386–405, https://doi.org/10.1016/j.jhydrol.2009.11.013, 2010.

2- Why do the authors select three flood volume dimensions, which are considered redundant? These variables are the same in nature and it is quite clear that the dependence between them should be high. The authors should explain why they do not select different variables representing different aspects of flow (with different nature) if they are really interested in applying a four-dimensional vine copula. Apparently, the whole process could be done using a bivariate copula. But if they are interested in developing a four-dimensional nonstationary-based vine copula, they should convince the readers why they select three of the dimensions from the same variable.

**Response:**

Thanks for this comment. In this paper, the flood series from the Xijiang River of China are chosen as the study data to illustrate the multivariate design methods under nonstationary conditions. According to the regulation for calculating design flood of China (Ministry of Water Resources of People's Republic of China, 1996), deriving flood hydrographs for hydraulic structures requires not only flood peak but also several flood volumes with different durations, such as 3 days, 7 days, 15 days and even 30 days (Xiao et al., 2009; Xiong et al., 2015; Li et al., 2017). Since this paper selects a hydrological design of China to perform the case study, the multivariate flood series should consist of flood peak and several flood volume variables with different durations. It is necessary to note that the proposed methods can be extended to other multivariate flood series, such as consisting of flood peak, flood volume and flood duration, which represent different aspects of flow.

In the revision, the above explanation has been added in the second paragraph of section 'Study area and data set' and the first paragraph of section 'methods'. The structure of the manuscript is also reorganized. The section 'Study area and data set' (i.e. section 3 in the

original manuscript) is listed as section 2, and the section 'methods' (i.e. section 2 in the original manuscript) is moved to section 3. Thus, the reasons why we select three flood volume dimensions have been explained before illustrating the methods. This adjustment would make it clear and logical for readers to understand the methods of this paper.

3- Why do the authors assume an exponential trend for the location parameters? How they make sure that there is not any other type of trend in the time series? What criteria is used to choose such an exponential trend? And why do not they use time dependent trend or any oscillation signal as covariates.

**Response:**

Thanks for this comment. In this paper, the domain of the location parameters (referring to the first moment or mean of flood series) should be $(0, +\infty)$. The exponential function enables the location parameters to always satisfy the domain and be meaningful, and this is something that some other functions such as linear and polynomial model are incapable of doing. In quite a few studies, the exponential function is selected to describe the nonstationarity of flow series (Vogel et al., 2011; Jiang et al., 2015; Read and Vogel, 2016; Yan et al., 2017). In addition, we also have employed a linear function to build the relationship between location parameters and explanatory covariates. The results suggest that the difference between the linear model and exponential model is very tiny. That is to say, the trend of the location parameter is mainly determined by the variations of the explanatory covariates, rather than the functions for describing the relationship between location parameter and covariates.

Based on cause-effect analysis, the flood processes of the Xijiang River are found to mainly be impacted by urbanization and reservoir operation. Therefore, reservoir index and urban population are definitely the more meaningful covariates than time variable in terms of mechanism of the flood nonstationarities. This is the reason why we use reservoir index and urban population as the explanatory covariates to describe the nonstationarity of location parameters rather than the time dependent trend. As we have explained above, the climatic conditions (such as extreme precipitation) of the study basin do not exhibit significant time variation in the observation period, and therefore the oscillation signals indicating the climatic nonstationary are not considered in this study.

In the revised manuscript, the above explanations have been added in the second paragraph of section 3.1.1 as follows:

Based on cause-effect analysis, the flood processes of the XRB were found to mainly be impacted by urbanization and reservoir operation. The reservoir index *RI* and urban population *Pop* were therefore used as candidate nonstationary indicators for the marginal distributions. Since the domain of location parameter $\mu$ is generally $(0, +\infty)$, $\mu$ was expressed as an exponential function of the covariates of *Pop* and *RI* to ensure a strictly positive mean of the flood series. A large number of studies used the exponential function to describe a nonstationary flow series (Vogel et al., 2011; Jiang et al., 2015; Read and Vogel, 2016; Yan et al., 2017).

4- The reason that the authors do not assume any time dependent dependency in roots T2 and T3 is not clear.

**Response:**

Thanks for this comment. It is true that the dependency (quantified by copula parameters) in roots T2 and T3 could be time dependent or nonstationary, at least theoretically. However, the estimations of the parameters in T2 and T3 depend on the estimated parameters in T1, and contain additional sources of uncertainty, particularly when the length of the observed flood series is limited. Therefore the present study kept the copula parameters in T2 and T3 constant to facilitate reliable parameter estimation. In the revision, we have added this reason in the fourth paragraph of section 3.1.2 as follows:

Theoretically, the copula parameters $\theta_{37|1,t}$ and $\theta_{315|1,t}$ in T2 and as well as $\theta_{715|13,t}$ in T3 could be nonstationary. However, the estimations of $\theta_{37|1,t}$, $\theta_{315|1,t}$ and $\theta_{715|13,t}$ depend on the estimated parameters in T1, and contain additional sources of uncertainty, particularly when the length of the observed flood series is limited. Therefore the present study kept the copula parameters in T2 and T3 constant to facilitate reliable parameter estimation.

In addition, this point is also stressed in the final paragraph of discussion section as below:

Nonstationary models generally improve performance in fitting observation data by reducing estimation bias (Jiang et al., 2015b), but this is usually achieved at the expense of increasing model complexity, such as adding more model parameters and introducing more nonstationary covariates, which might induce additional sources of model uncertainty (Serinaldi and Kilsby, 2015; Read and Vogel, 2016). The nonstationarities in the present study for both higher-order marginal distribution parameters and copula parameters in roots T2 and T3 were ignored to avoid possible larger uncertainty in parameter estimation considering the limited length of the flood series.

5- Why do not the authors select the copula in equation 6 from the extreme copula families. And why do not they use any Goodness of Fit Test to select the best fitted copula from different copula families?

**Response:**

In this study, we employ the Gumbel-Hougaard copula (as expressed by Eq. 6), which is an extreme-value copula and widely used in hydrology, to construct the dynamic C-vine copula. In flood frequency analysis, the upper tail of the flood distribution deserves more attention because it allows the quantification of risks of the more serious flood events. The Gumbel-Hougaard copula accounts for the upper tail dependence, and is well-suited to the dependence structure of a multivariate flood distribution (Salvadori et al., 2007; Zhang and Singh, 2007; Xiong et al., 2015). In addition, we also have considered more copulas used in hydrology, such as Frank, Clayton and *t* student copulas. The results indicate that Gumbel-Hougaard copula has the best fitting quality.

The reason why the Gumbel-Hougaard copula is selected has been added in the fifth paragraph of section 3.1.2 as follows:

In flood frequency analysis, the upper tail of the flood distribution deserves more attention because it allows the quantification of risks of the more serious flood events. The Gumbel-Hougaard copula, an extreme-value copula widely used in hydrology, accounts for the upper

tail dependence, and is well-suited to the dependence structure of a multivariate flood distribution (Salvadori et al., 2007; Zhang and Singh, 2007; Xiong et al., 2015). Consequently, the present study employed the bivariate Gumbel-Hougaard copula to construct the dynamic C-vine copula formulated by Eq. (4).

6- In equation 7, it is not clear that what is the covariate? Is time is the covariate? And again, why the authors use an exponential nonlinear trend to express the non-stationarity in the copula parameter? What if a linear or polynomial regression model is fitted well to express the trend in the copula parameters.

**Response:**

The covariate in Eq. (7) denotes the explanatory variable for describing the nonstationarity of copula parameter. Based on cause-effect analysis for nonstationarity of the flood series, the covariates used in this study contain urban population (*Pop*) and reservoir index (*RI*), both of which are factors altering the natural flood processes. In the revision, the covariates in Eq. (7) have been specified as *Pop* and *RI* (please see below). Since time variable has no cause-effect relationship with nonstationarity of the flood series, it is not used as the covariate of copula parameter.

In this study, we select Gumbel-Hougaard copula to construct the joint distribution of multivariate flood series, and the domain of the copula parameter is $[1,+\infty)$. To satisfy the domain range of the copula parameter under any condition, an exponential model expressed as $\theta_{c,t} = 1 + \exp(\cdot)$ is employed to build the relationship between copula parameter and covariates.

A linear or polynomial regression model could induce the copula parameter beyond the domain and to be meaningless.

In the revision, the above explanation has been added in the fifth paragraph of section 3.1.2 as follows:

To satisfy the domain range of the copula parameter under any condition, the copula parameter $\theta_c$ was written as the sum of one and an exponential function of the covariates.

Similar to the marginal distributions, the four candidate models of time-varying dependence were formulated as follows:

$$\begin{aligned}
\theta_{c,t} &= 1 + \exp\left(\beta_0 + \beta_1 Pop_t + \beta_2 RI_t\right) \\
\theta_{c,t} &= 1 + \exp\left(\beta_0 + \beta_1 Pop_t\right) \\
\theta_{c,t} &= 1 + \exp\left(\beta_0 + \beta_1 RI_t\right) \\
\theta_{c,t} &= 1 + \exp\left(\beta_0\right)
\end{aligned} \tag{7}$$

7- The authors talk about robustness of their model in lines 217 and 241. What is the definition of the robustness for these two cases?

**Response:**

The so-called robustness suggests the uncertainty or reliability of the estimated parameters. In the revision we have used the more suitable words of uncertainty and reliability. In addition, It should be noted that the sentence in line 217 of the original manuscript have been deleted to make the manuscript succinct.

8- The authors have not done any uncertainty analysis for estimation of the marginal and copula parameters through time.

**Response:**

Thanks for this constructive comment. In the revision, we have performed an uncertainty analysis for the estimated marginal and copula parameters using the parametric bootstrap method (Kyselý, 2009). The 95% uncertainty intervals of the estimated parameters are shown in Tables 2 and 3 as follows:

**Table 2** Results of nonstationary analysis for the marginal distributions of $(Q_1, V_3, V_7, V_{15})$

| Flood variable | Distribution | $\mu$ | | | $\sigma$ | $\nu$ | $p\_$KS |
|---|---|---|---|---|---|---|---|
| | | $\alpha_0$ | $\alpha_1$ | $\alpha_2$ | | | |
| $Q_1$ | GEV | 10.050*** | 0.0212*** | −2.166** | 6892.085*** | −0.271** | 0.713 |
| | | [9.931, 10.182] | [0.005,0.036] | [−4.006,−0.481] | [5313.291, 8176.206] | [−0.527, −0.092] | |
| $V_3$ | Gamma | 1.866*** | 0.0185** | −2.094** | 0.261*** | - | 0.832 |
| | | [1.751, 1.977] | [0.002, 0.034] | [−3.801, −0.403] | [0.209, 0.300] | | |
| $V_7$ | Gamma | 2.638*** | 0.0119** | −1.934** | 0.269*** | - | 0.907 |
| | | [2.522,2.754] | [−0.005,0.028] | [−3.713,−0.166] | [0.215,0.308] | | |
| $V_{15}$ | Gamma | 3.258*** | - | −1.525** | 0.265*** | - | 0.926 |
| | | [3.213,3.354] | | [−2.807,0.155] | [0.215,0.307] | | |

$\alpha_1$ and $\alpha_2$ are the parameters related to urban population (*Pop*) and reservoir index (*RI*), respectively. The symbols '***', '**' and '*' denote that the estimated model parameters are significant at the levels of 0.01, 0.05 and 0.1, respectively. The numbers in brackets are the 95% uncertainty interval. $p\_$KS stands for the *p*-value of the KS test for marginal distributions.

**Table 3** Results of nonstationary analysis for the dependence structure of $(Q_1, V_3, V_7, V_{15})$

| Copula parameter | Model parameters | | |
|---|---|---|---|
| | $\beta_0$ | $\beta_1$ | $\beta_2$ |
| $\theta_{13}$ | 3.023*** | - | - |
| | [2.816, 3.249] | | |
| $\theta_{17}$ | 1.719*** | - | - |
| | [1.483, 1.976] | | |
| $\theta_{115}$ | 1.461*** | −0.111** | 9.426** |
| | [0.958,2.038] | [−0.021,−0.226] | [0.970,20.416] |
| $\theta_{37\|1}$ | 0.0926* | | - |
| | [−0.316, 0.473] | | |
| $\theta_{315\|1}$ | −1.444** | - | - |
| | [−3.036,−0.693] | | |
| $\theta_{715\|13}$ | −0.231* | - | - |
| | [−0.728, 0.199] | | |

$\beta_1$ and $\beta_2$ are the parameters related to urban population (*Pop*) and reservoir index (*RI*), respectively. The symbols '***', '**' and '*' denote that the estimated model parameters are significant at the levels of 0.01, 0.05 and 0.1, respectively. The numbers in brackets are the 95% uncertainty interval. *p*_PIT stands for *p*-value of the PIT test for the C-vine copula.

**Newly cited reference:**

Kyselý, J.: A cautionary note on the use of nonparametric bootstrap for estimating uncertainties in extreme-value models, Journal of Applied Meteorology & Climatology, 47 (12), 3236-3251, 2009.

9- Section 4.1 and 4.2 should move to the methodology, as they are not related to the results section.

**Response:**

In the revision, the context about methodology in section 4.1 and 4.2 has been merged in the methods section.

10- Finally, the manuscript would greatly benefit from the input of a native English speaker.

**Response:**

The revised manuscript have been proofed by a native English speaker.

[revised manuscript text omitted]

---

## Referee Report (RR1)

**REVIEW REPORT 2**

**Journal:** Hydrology and Earth System Sciences
**Paper:** hess-2018-291
**Title:** Multivariate hydrologic design methods under nonstationary conditions and application to engineering practice
**Author(s):** Cong Jiang, Lihua Xiong, Lei Yan, Jianfan Dong and Chong-Yu Xu

**GENERAL COMMENTS.**

It is rather difficult to evaluate this work. The Authors provided some answers to the issues raised by the Referees, but, in my opinion, these replies are also questionable. The Editor will take a decision. Some comments follow below.

**SPECIFIC COMMENTS.**

**Page(s) 9, Line(s) 215–216.**

> **Author(s).** Given the limited length of the flood series used in this study, the higher-order distribution parameters such as scale and shape parameters were fixed to avoid possible large uncertainty.

> **Referee.** However, such an assumption can make the work rather weak...

**Page(s) 10, Line(s) 219–220.**

> **Author(s).** Since the domain of location parameter $\mu$ is generally $(0, +\infty)$, $\mu$ was expressed as an exponential function...

> **Referee.** If this is the problem, any polynomial of the form $g(t) = t^{2k}$, with integer $k > 0$, would generally satisfy the non-negativity constraint...

**Page(s) 10, Eq(s) 3.**

> In Eq. (3) the Authors are implicitly assuming a "multiplicative" model, being the exponential of a linear combination of the arguments: any reason for doing that?

**Page(s) 12, Line(s) 273.**

> **Author(s).** Therefore the present study kept the copula parameters in T2 and T3 constant...

> **Referee.** However, as above, such an assumption can make the work rather weak...

**Page(s) 13, Line(s) 284–285.**

> **Author(s).** To satisfy the domain range of the copula parameter under any condition, the copula parameter $\theta_c$ was written as the sum of one and an exponential function of the covariates.

> **Referee.** As above, if this is the problem, any polynomial of the form $g(t) = 1 + t^{2k}$, with integer $k > 0$, would generally satisfy the constraint...

**Page(s) 13, Line(s) 296–297.**

> **Author(s).** The best nonstationary model for each pair copula in T1 was chosen from the nonstationary models generally expressed by Eq. (7) in terms of the AIC value (Akaike, 1974).

> **Referee.** Maybe, it would be better to use a Corrected-AIC procedure: it should account for possible over-parametrization...

**Page(s) 14, Line(s) 317–318.**

**Author(s).** OR, AND and Kendall cases...

**Referee.** These were first introduced, and theoretically discussed, in Salvadori and De Michele (2004): please fix the references (always give proper credits to whom deserve credits).

**Page(s) 15, Eq(s) 10.**

The exceedance probability in Eq. (10) can be calculated directly via Eq. (1) in Salvadori et al. (2013), exploiting the inclusion-exclusion principle.

**Page(s) 20, Line(s) 448–450.**

**Author(s).** In addition, the change-point detection method based on the Cramer-von Mises statistic (Bucher et al., 2014) was employed to detect possible nonstationarities in both the marginal distributions and dependence of the multivariate flood series...

**Referee.** Please show the p-values of the tests.

**References**

Salvadori, G., De Michele, C., 2004. Frequency analysis via copulas: theoretical aspects and applications to hydrological events. Water Resour. Res. 40, W12511, doi: 10.1029/2004WR003133.

Salvadori, G., Durante, F., De Michele, C., 2013. Multivariate return period calculation via survival functions. Water Resour. Res. 49, 2308–2311, doi: 10.1002/wrcr.20204.

---

## Author Response (AR2)

**Cover letter for hess-2018-291**

**Dear Editor,**

On behalf of my co-authors, I would like to express our great appreciation to you and the reviewers for the efforts on reviewing our manuscript entitled "Multivariate hydrologic design methods under nonstationary conditions and application to engineering practice" (ID: hess-2018-291).

All the comments raised by the reviewers are very professional and have been addressed in the revision of the manuscript. A point-by-point response to the comments is presented as appendix to this letter. Attached please find the revised manuscript with all revisions marked in red color. We hope the revised manuscript will meet with the approval for publication in HESS.

Great thanks to you and reviewers once again for the efforts on our paper during these two review circles.

With all best wishes

Yours

Cong JIANG

On behalf of my co-authors
* * *
Cong JIANG, PhD
School of Environmental Studies
China University of Geosciences
Wuhan 430074
People's Republic of China
E-mail: jiangcong@cug.edu.cn
Telephone: +86-13659831242

**Reply to Referee #1**

**General comments**

It is rather difficult to evaluate this work. The Authors provided some answers to the issues raised by the Referees, but, in my opinion, these replies are also questionable. The Editor will take a decision. Some comments follow below.

**Response:**

   Great thanks for your professional and insightful comments on this manuscript in two review circles. All your comments are of great significance for improving the quality of this manuscript, and have been addressed in revising the manuscript. Please see our point-to-point replies below.

**Specific comments**

**(1)  Page(s) 9, Line(s) 215–216.**

   **Author(s).** Given the limited length of the flood series used in this study, the higher-order distribution parameters such as scale and shape parameters were fixed to avoid possible large uncertainty.

   **Referee.** However, such an assumption can make the work rather weak. . .

**Response:**

   In the revised manuscript, this assumption has been removed, and the nonstationarities of the higher-order distribution parameters such as scale and shape parameters have been examined by the time-varying moments model. The results indicate that both the scale and shape parameters for all flood variables in this study are stationary. Thus this modification does not make any difference to the results in the previous version of the manuscript.

**(2)  Page(s) 10, Line(s) 219–220.**

   **Author(s).** Since the domain of location parameter $\mu$ is generally $(0; +1)$, $\mu$ was expressed as an exponential function. . .

   **Referee.** If this is the problem, any polynomial of the form $g(t) = t^{2k}$, with integer $k > 0$, would generally satisfy the non-negativity constraint. . .

**Response:**

   Thanks for this insightful comment. It is really true that the domains of distribution parameters could not be a strong argument for the selection of the functions expressing the relationships between distribution parameters and covariates. Choosing a proper function to model the nonstationarities of hydrological distribution parameters appears to be an issue common to most current relevant studies, since it is hard to provide some theoretical supports beyond the fitting quality. Polynomial functions are able to generally satisfy the non-negativity constraint, but seem to be too complex and might result in overfitting of distribution parameters when the model contains multiple explanatory covariates. In previous literatures, linear and exponential functions have been widely used to build the relationships between distribution parameters and covariates (Strupczewski et al., 2001; Vogel et al., 2011; Jiang et al., 2015; Read and Vogel, 2016; Yan et al., 2017). In the revision, in addition to exponential function, linear function has also been supplemented as another candidate model for building the relationships between distribution parameters and covariates, so that both linear and nonlinear relationships are considered in this

study. The proper model is chosen from these two types of functions in terms of fitting quality measured by corrected AIC. The results indicate that exponential function performs slightly better than linear model in characterizing the nonstationarities of distribution parameters. In the revised manuscript, Eq. (3) has been modified as:

$$
\begin{aligned}
&\text{Linear:} && \mu_t = \alpha_0 + \alpha_1 Pop_t + \alpha_2 RI_t \\
&\text{Exponential:} && \mu_t = \exp\left(\alpha_0 + \alpha_1 Pop_t + \alpha_2 RI_t\right)
\end{aligned}
\tag{3}
$$

**(3) Page(s) 10, Eq(s) 3.**

In Eq. (3) the Authors are implicitly assuming a "multiplicative" model, being the exponential of a linear combination of the arguments: any reason for doing that?

**Response:**

Thanks for this comment. In the original manuscript, Eq. (3) gives a "multiplicative" model, which is able to take into account the possible interaction between different covariates leading to the flood nonstationarity. In the revised manuscript, we have added this explanation in the second paragraph of Section 3.1.1 as follows:

As above, the linear expression in Eq. (3) gives an additive model which suggests that the effects of the covariates *RI* and *Pop* on $\mu$ are independent, while the exponential expression defines a multiplicative model which is able to take into account the possible interaction between the covariates *RI* and *Pop*.

**(4) Page(s) 12, Line(s) 273.**

**Author(s).** Therefore the present study kept the copula parameters in T2 and T3 constant. . .

**Referee.** However, as above, such an assumption can make the work rather weak. . .

**Response:**

In the revised manuscript, this assumption has been removed and the nonstationarities of the copula parameters in T2 and T3 are considered. The results indicate that all these copula parameters in T2 and T3 are stationary. This finding indicates that this assumption does not make any difference to the results in the previous version of the manuscript.

**(5) Page(s) 13, Line(s) 284–285.**

**Author(s).** To satisfy the domain range of the copula parameter under any condition, the copula parameter $\theta_c$ was written as the sum of one and an exponential function of the covariates.

**Referee.** As above, if this is the problem, any polynomial of the form $g(t) = 1 + t^{2k}$, with integer $k > 0$, would generally satisfy the constraint. . .

**Response:**

In the revised manuscript, we have supplemented a linear function as a candidate model for describing the nonstationarity of the copula parameter. The proper model is chosen from linear and exponential functions in terms of fitting quality measured by corrected AIC. For more detailed explanation, please also refers to the Response in the above made to Comment (2).

**(6) Page(s) 13, Line(s) 296–297.**

**Author(s).** The best nonstationary model for each pair copula in T1 was chosen from the nonstationary models generally expressed by Eq. (7) in terms of the AIC value (Akaike, 1974).

**Referee.** Maybe, it would be better to use a Corrected-AIC procedure: it should account for possible over-parameterization. . .

**Response:**

In the revision, corrected-AIC (Hurvich and Tsai, 1989) has been used as the criterion to perform model selection. It is found that corrected-AIC yields the same model selection results as with AIC.

**Newly cited reference:**

Hurvich, C. M., and Tsai, C. L.: Regression and time series model selection in small samples, Biometrika, 76, 297–307, 1989.

**(7) Page(s) 14, Line(s) 317–318.**

**Author(s).** OR, AND and Kendall cases. . .

**Referee.** These were first introduced, and theoretically discussed, in Salvadori and De Michele (2004): please fix the references (always give proper credits to whom deserve credits).

**Response:**

In the revision, we have added this citation.

**(8) Page(s) 15, Eq(s) 10.**

The exceedance probability in Eq. (10) can be calculated directly via Eq. (1) in Salvadori et al. (2013), exploiting the inclusion-exclusion principle.

**Response:**

Eq. (10) is the theoretical expression of the AND exceedance probability. The value of Eq. (10) is calculated by the Monte Carol method, which is similar to the method in Salvadori et al. (2013). In the revised manuscript, we have added this citation in the final paragraph of Section 3.2.2.

**(9) Page(s) 20, Line(s) 448–450.**

**Author(s).** In addition, the change-point detection method based on the Cramer-von Mises statistic (Bucher et al., 2014) was employed to detect possible nonstationarities in both the marginal distributions and dependence of the multivariate flood series. . .

**Referee.** Please show the p-values of the tests.

**Response:**

In the revised manuscript, we have added a table to display the *p*-values of the change-point test.

**Table 4** Results of change-point detection for the marginal distributions and dependence of

$(Q_1, V_3, V_7, V_{15})$

| Flood series | Change point of margin | *p*-value | Flood series | Change point of dependence | *p*-value |
|---|---|---|---|---|---|
| $Q_1$ | 1993 | 0.072 | $(Q_1, V_3)$ | 1955 | 0.083 |
| $V_3$ | 1993 | 0.186 | $(Q_1, V_7)$ | 1955 | 0.537 |
| $V_7$ | 1994 | 0.752 | $(Q_1, V_{15})$ | 1972 | 0.599 |
| $V_{15}$ | 1981 | 0.423 | $(Q_1, V_3, V_7, V_{15})$ | 1972 | 0.995 |

**Reply to Referee #4**

**General comments**

Overall, this paper is well written and is of great interest to the flooding research community. Considering the floods in the context of multiple variables is important as such cases have been often observed in practice. I would like to invite the authors to address the following comments before this paper can be accepted.

**Response:**

We appreciate your positive evaluation as well as constructive comments on this paper. All of your concerns have been addressed in the manuscript. Please see our point-to-point reply below.

**Specific comments**

(1) While the paper focuses on the multivariate, it actually simplifies it through decomposing the multivariate into many bivariate cases. I understand this is in the consideration for parameter estimation. However, this also may result in possible bias in the results. I suggest the authors to give some discussions on this.

**Response:**

Thanks for this constructive comment. In this study, we constructed the joint distribution of the multivariate flood variables through decomposing the multivariate dependence into many bivariate pair copulas. To make it easy in parameter estimation, the model parameters for each pair copula were separately estimated. It is worth noting that these parameters can be also simultaneously estimated. These two methods could result in possible difference in parameter estimation. In the revision, we have added this discussion in the fifth paragraph of Section 3.1.2 as follows:

To make it easy in parameter estimation, the model parameters for each pair copula were separately estimated. The model parameters for $\theta_{13,t}$, $\theta_{17,t}$ and $\theta_{115,t}$ in T1 were first estimated, and those for the remaining copula parameters $\theta_{37|1,t}$, $\theta_{315|1,t}$ and $\theta_{715|13,t}$ in T2 and T3 were then estimated in sequence. It is worth noting that these parameters can be also simultaneously estimated. These two methods could result in possible difference in parameter estimation.

(2) Can the authors please present some statistical figures to pair the $Q$ and $V$ (just use the original data to see whether there is a strong correlation). I think these plots would be very helpful.

**Response:**

In the revised manuscript, we have added a figure to present statistical correlations between flood peak and flood volumes as follows:

[Figure]

**Figure 5.** Statistical correlations between flood peak and flood volumes

(3) "In reality, all parameters of the flood distribution can be nonstationary, but this paper only considered nonstationarity of the location parameter (referring to the first moment or mean of the flood series)." This is an important assumption that needs more explanation. Does this assumption will significant affect the results? Have previous studies made similar assumptions?

**Response:**

In accordance with Comment (1) by Referee # 1, this assumption has been removed, and the nonstationarities of all distribution parameters are taken into consideration in the revision. In reality, all parameters of the flood distribution could be nonstationary, therefore this assumption could has some impacts on the results. As for this case study, even after the nonstationarity scenarios of all three distribution parameters has been considered, only the location parameter is found to be nonstationary and the higher-order distribution parameters are stationary, which does not make any difference to our previous results. Sometimes, this assumption that only the location parameter of hydrological distributions is considered to be nonstationary can be found in some previous studies (Sarhadi et al., 2016), where the higher-order distribution parameters were treated as constant.

(4) Why annual maximum daily discharge, annual maximum 3-day flood volume, annual maximum 7-day flood volume, and annual maximum 15-day flood volume are used as the multivariate flood series. What are the physical significances of selecting such variables? Do they link to the catchment properties?

**Response:**

The calculation of design floods in China involving the derivation of flood hydrographs for hydraulic structures requires not only the flood peak, but also flood volumes with different durations, such as 3 days, 7 days, 15 days and 30 days. The selection of flood variables is associated with the durations of flood processes (or flood hydrographs). For a large catchment such as the Xijiang River basin, the duration of a flood process is usually longer than ten days. Therefore, the annual maximum daily discharge, annual maximum 3-day flood volume, annual maximum 7-day flood volume, and annual maximum 15-day flood volume are used to constitute the multivariate flood variables for deriving the design flood hydrograph. In the revised manuscript, we have added this explanation in the second paragraph of Section 2.

(5) What does ARR mean in the Abstract? Should give the full name as it appeals at the first time.

**Response:**

Thanks for this comment. ARR is a typo and should be AAR (average annual reliability). In the revision, we have made this modification.

(6) Authors have mentioned two remarks in the end of the paper, of which the first one is related to the observations that are used for the flooding risk analysis. I totally agree with the authors in that the data with sufficient length is critical to enable the statistical analysis, which has been systematically analyzed in Zheng et al. (2018). I suggest the authors to have a look at this paper that focus on the data collecting methods for rainfall and floods.

**Response:**

Thanks for recommending this suitable reference, which is helpful supporting our viewpoint.

In the revised manuscript, we have added this citation in the fifth paragraph of Section 5.

**Newly cited reference:**

[revised manuscript text omitted]

---

## Author Response (AR3)

**Cover letter for hess-2018-291**

**Dear Editor,**

On behalf of my co-authors, I would like to express our great appreciation to you and the reviewers for reviewing our manuscript entitled "Multivariate hydrologic design methods under nonstationary conditions and application to engineering practice" (ID: hess-2018-291). According to your suggestion, we have added these two important references in the revised manuscript.

Great thanks to you once again for the efforts on our paper.

With all best wishes

Yours

Cong JIANG

On behalf of my co-authors
* * *
Cong JIANG, PhD

[revised manuscript text omitted]